# Coherent photoelectrical readout of single spins in silicon carbide at room temperature

Tetsuri Nishikawa [1,7], Naoya Morioka [1,2,7] ✉, Hiroshi Abe [3], Koichi Murata [4], Kazuki Okajima [1], Takeshi Ohshima [3,5], Hidekazu Tsuchida [4] & Norikazu Mizuochi [1,2,6] ✉

Establishing a robust and integratable quantum system capable of sensitive qubit readout at ambient conditions is a key challenge for developing prevalent quantum technologies, including quantum networks and quantum sensing. Paramagnetic colour centres in wide bandgap semiconductors provide optical single-spin detection, yet realising efficient electrical readout technology in scalable material will unchain developing integrated ambient quantum electronics. Here, we demonstrate photoelectrical detection of single spins in silicon carbide, a material amenable to large-scale processing and electronic integration. With efficient photocarrier collection, we achieve a 1.7–2 times better signal-to-noise ratio for single spins of silicon vacancies with electrical detection than with optical detection suffering from saturating fluorescence and internal reflection. Based on our photoionisation dynamics study, further improvement would be expected with enhanced ionisation. We also observe single-defect-like features in the photocurrent image where photoluminescence is absent in the spectrum range of silicon vacancies. The efficient electrical readout in the mature material platform holds promise for developing integrated quantum devices.

Developing scalable quantum systems operating at room temperature and providing easy state readout is a key challenge in modern science and technology toward various quantum technologies, including advanced sensing[1] and quantum network[2]. Semiconductor-based qubits with electrical readout capability are an attractive platform for miniaturised and portable quantum devices due to their mature integration technology and compatibility with peripheral circuitry[3,4]. The leading systems, such as quantum dots[5,6], electron spins[7] of donors, and nuclear spins[8], have successfully implemented electrical readout at low temperatures. However, the cryostat restricts the system's volume and portability, which has led to the search for a material platform and technologies that enable the electrical detection of single qubits at ambient conditions. The photoelectrically detected magnetic resonance (PDMR) technique[9], first developed in diamond and silicon carbide (SiC) for coherent detection of spin ensembles of colour centres[9,10] and surrounding nuclei[11,12], has achieved a sensitivity of single-spin detection at room temperature only for a nitrogen-vacancy (NV) centre in diamond[13,14], a spin-active colour centre. PDMR is a spin-to-charge conversion process that utilises photoionisation with spin-dependent probability due to the defect's internal spin dynamics, making the spin state detectable by photocurrent measurement (Fig. 1a).

As a practical consideration, the quantum system is preferably hosted in a technology-friendly material. Therefore, it is crucial to provide a more technology-friendly and wafer-scale material platform capable of room-temperature electrical single-spin readout that is not

[1]Institute for Chemical Research, Kyoto University, Uji, Japan. [2]Center for Spintronics Research Network, Institute for Chemical Research, Kyoto University, Uji, Japan. [3]National Institutes for Quantum Science and Technology (QST), Takasaki, Japan. [4]Central Research Institute of Electric Power Industry, Yokosuka, Japan. [5]Department of Materials Science, Tohoku University, Sendai, Japan. [6]International Center for Quantum-field Measurement Systems for Studies of the Universe and Particles (QUP), KEK, Tsukuba, Japan. [7]These authors contributed equally: Tetsuri Nishikawa, Naoya Morioka. ✉e-mail: morioka.naoya.8j@kyoto-u.ac.jp; mizuochi@scl.kyoto-u.ac.jp

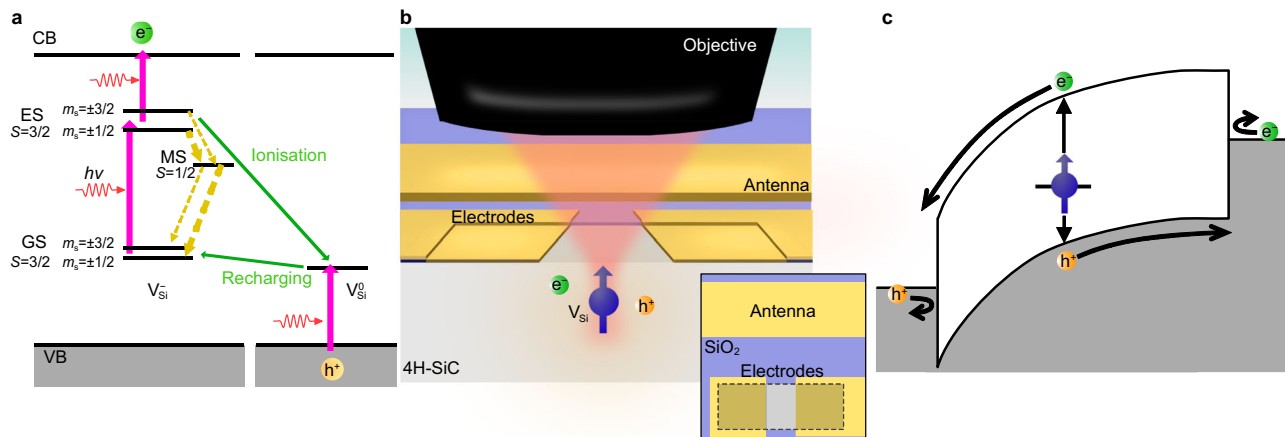

**Fig. 1 | Schematics of the experiments. a** Energy diagram for PDMR process of silicon vacancy in 4H-SiC. The vertical pink and the slanted green arrows illustrate the optical excitation from the ground state (GS) to the excited state (ES) and the optical charge-state transition via the neutral state ($V_{Si}^0$) accompanied by carrier generation in the conduction band (CB) and the valence band (VB). The yellow broken arrows indicate the intersystem crossing via the metastable state (MS), and their thickness depicts transition rates. **b** Schematic of PDMR device composed of gold Schottky contacts and RF antenna (gold) and SiO$_2$ isolation (blue) on 4H-SiC (grey). The inset at the bottom right shows the vertical view of the device. **c** Band diagram of the metal-semiconductor-metal PDMR device biased at the flat-band condition.

limited to diamond NV centres. Further, the readout technique must achieve a high signal-to-noise ratio (SNR). Electrical charges can be collected easily under an electrical field, in contrast to the difficulty in photon collection from high-refractive-index materials[15]. Indeed, the number of collected charges from a single NV centre exceeded the photon counts[13], indicating the potential to achieve a higher SNR[16] than with the optical detection of magnetic resonance (ODMR), a commonly used spin-detection method for colour centres. However, in actual experiments, electrical noise easily couples to the detection system and limits SNR, whereas ODMR can achieve the photon shot-noise limit. For this reason, the achievement of high spin-detection SNR in PDMR has been a challenging task.

Here, we report room-temperature electrical detection of single electron spins in SiC, a wide-bandgap semiconductor for new-generation power devices. Our target is a negatively charged silicon vacancy in 4H polytype SiC, a spin 3/2 system[17] that exhibits a long spin-coherence time even at room temperature[18] and is actively studied for quantum sensing[19,20] and quantum communication[21,22]. SiC's mature device technologies and large-scale wafer synthesis capability make it a prominent candidate for integrated quantum devices. Compared with diamond, achieving room-temperature electrical single-spin detection in SiC is challenging due to its high electrical conductivity, even in pure materials with ppb-level impurity concentrations due to the shallow donor and acceptor levels in SiC. We solve this problem by the design of the device structure and the operation conditions. We experimentally demonstrate single-spin detection with a superior SNR by PDMR compared to that by ODMR. Also, theoretical modelling indicates a potential for further improvement of the SNR with PDMR at increased ionisation efficiency. These results demonstrate that the electrical readout of qubits in SiC provides a significant step for realising room-temperature quantum electronics.

## Results
### Photoelectrical imaging and spin detection of single defect in silicon carbide
We use an epitaxially grown 4H-SiC sample in all experiments. Isolated silicon vacancies are created by electron-beam irradiation and annealing. A PDMR device with electrodes for photocurrent collection and an antenna for radio-frequency waves (RF) for spin control was fabricated (Fig. 1b and Methods). Our device is a metal-semiconductor-metal (MSM) Schottky diode fabricated with gold contacts on 4H-SiC.

Although the device structure is similar to those used in previous studies for NV centres in diamond, the previous studies targeted Ohmic contacts by applying annealing[9,13]. In contrast, we intentionally avoided forming Ohmic contacts to achieve both high carrier collection efficiency and low leakage current. Therefore, the design strategy in this study is different from the previous studies. It is of significant importance to note that silicon carbide exhibits considerably lower resistivity at room temperature compared to diamond, the material that previously demonstrated room-temperature electrical single-spin detection. High-purity synthetic diamond is nearly insulating with a resistivity as high as ~$10^{15}$ Ωcm[23] due to the deep donor level of the nitrogen impurity. In contrast, silicon carbide with a low dopant concentration at about $10^{13}$–$10^{15}$ cm$^{-3}$ has a resistivity of about 1–$10^4$ Ωcm[24] due to shallow donor and acceptor levels. In the case of the material used in electrical ensemble spin detection containing a high-density ensemble of defects[10,12], the generated defects trap the charges and make the material semi-insulating[25]. Nevertheless, nanoampere-level dark current was reported in the device for the previous demonstration of ensemble spin PDMR in SiC[10]. It is, therefore, necessary to configure the device in such a way that it efficiently blocks the leakage current. Conversely, the seemingly contradictory requirement for highly efficient photocurrent collection is simultaneously fulfilled for the purpose of single defect detection. The Ohmic contact is, however, not required to achieve efficient photocurrent collection. MSM Schottky devices can achieve carrier collection efficiency close to 100% when the device is operated above the flat-band condition above the flat-band voltage[26]. The schematic band diagram[27] of the MSM Schottky device at the flat-band condition is shown in Fig. 1c. The flat-band condition is achieved by applying the bias voltage to the MSM Schottky device slightly above the full-depletion condition. In the case of our p-type material, above the flat-band condition, the potential barrier becomes zero for photogenerated holes that travel to the negatively biased contact. The energy band at the other contact is bent by the reverse bias, but this band bending is beneficial for electrons to be collected by the electrode, again with no potential barrier. Concurrently, the charge injection from the metal to the semiconductor is impeded by the Schottky barrier at the interface, which is advantageous in reducing the dark current that contributes to background noise. The use of Schottky contacts eliminates the need for ohmic annealing and allows electrode formation without affecting the characteristics of defects. It is also advantageous in that it allows a wide variety of electrode options to achieve leakage current reduction. The

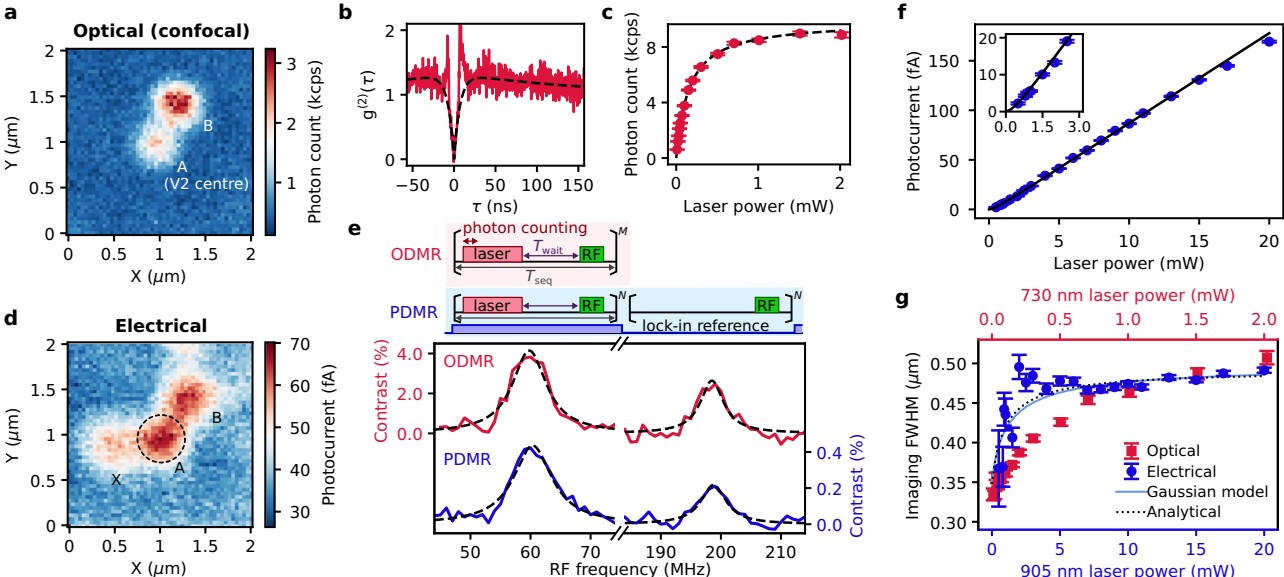

**Fig. 2 | Electrical detection of a single V2 centre in 4H-SiC. a** Fluorescence scanning image of 2 μm square area between electrodes by 730 nm laser at 30 μW. The integration time is 50 ms per pixel. **b** The photon autocorrelation function $g^2(\tau)$ of defect A. The red solid line and black broken line are background-corrected[68] experimental data and fitting, respectively. **c** Laser-power dependence of the fluorescence for defect A. The fit function (black broken line) is $I_0 P / (P_0 + P)$ where $P$ is the power with $I_0 = 9.9$ kcps and $P_0 = 155$ μW. **d** Photocurrent image at the same area as **a** with 905 nm laser at 4 mW. The applied voltage is 10 V. The data sampling interval is 70 ms per pixel. **e** The sequences and spectra of ODMR (red line) and PDMR (blue line) with RF sweep under a static field. The linear background component is subtracted from the data. The black dashed lines are Lorentzian fits. ODMR is measured with 1 μs laser pulses at 500 μW and 140 ns RF pulses at 18.2 dBm. PDMR is measured with 1 μs laser pulses at 20 mW modulated at 7 Hz and the same RF conditions for the ODMR. The approximately 140 MHz separation of two peaks indicates a single V2 centre[37]. **f** Laser-power dependence of the photocurrent of defect A. The fit function (black line) is $\alpha P^2 / (1 + P/P_0)$. The inset shows the enlarged view of the low laser power region. **g** Laser power dependence of the FWHM spot size in optical and electrical imaging measurements for defect A. The solid line is the calculated FWHM by fitting the photocurrent profile model with a Gaussian function, and the dotted line is the analytical FWHM model. All error bars in this figure show ±1 standard error.

gold used in this study exhibits high barrier heights for both p-type (1.35 eV[28]) and n-type (1.73 eV[29]) 4H-SiC, which is effective in blocking the injection of both electrons and holes at two MSM contacts into SiC. Furthermore, to minimise the leakage current, the Schottky contact area is designed to be relatively small (approximately 120 μm²). Additionally, the bonding pads and wiring patterns are insulated from SiC by a 200 nm thick SiO₂ layer to prevent unwanted electrical connections. We design the device to achieve the flat-band condition at about 9 V, and the achieved dark current is about 10 fA at the operating voltage of 10 V, sufficiently small to detect the signal from single defects (Supplementary Note 1).

To achieve high charge collection efficiency, the charge capture by defects in the channel is important. Recently, studies on the capture of charge emitted from single defects have been reported[30,31]. Since we generate single silicon vacancies by electron irradiation, carbon vacancies, a lifetime killer in silicon carbide[32], are also generated in the sample with a density higher than silicon vacancies. Therefore, carbon vacancies are expected to be the dominant charge capture centre in our device. The expected density of the carbon vacancy is about $1 \times 10^{13}$ cm⁻³ considering the estimated as-grown density ($2 \times 10^{13}$ cm⁻³) and created density by the electron-beam irradiation ($1 \times 10^{13}$ cm⁻³)[33]. With this density, the carrier lifetime is expected to be about 1 μs[32,34], which is much longer than the expected carrier transit time on the order of nanoseconds. Although the density of the dopants is higher than that of the carbon vacancy, the emission from shallow impurity levels is fast in silicon carbide. Also, as we operate the device above the flat-band condition, the dopants are expected to be fully depleted. Therefore, we expect the effect of the charge capture on the charge collection to be insignificant, and thus, high charge collection efficiency is expected.

We characterise the area between the electrodes optically and photoelectrically. We excite the colour centres with a 730 nm laser and detect photons above 900 nm for optical characterisation to maximise

the excitation efficiency for silicon vacancies[35] while suppressing Raman-scattered background light. The fluorescence image shown in Fig. 2a shows isolated colour centres. We identify defect A with $g^2(0) < 0.5$ (Fig. 2b) and a saturation fluorescence intensity of 9.9 kilocounts/s (kcps) (Fig. 2c) as a single silicon vacancy at the quasi-cubic site[36] (V2 centre) by ODMR with the characteristic peaks separated by ~140 MHz[37] shown in Fig. 2e. The observed saturation fluorescent intensity is comparable to the reported values of single V2[18,38,39]. The nearby defect B is another single colour centre without any ODMR peak, assumed to be a V1 centre, another silicon vacancy at the hexagonal site[36].

When we measure a photoelectrical image with the 730 nm laser, isolated defects are not observed, probably due to a large background from the interband two-photon electron-hole pair generation (bandgap: 3.24 eV) and ionisation of defects other than V2 (Supplementary Note 2). We successfully obtain a clean single-defect-like photoelectrical image with a 905 nm laser, close to the zero-phonon line of the V2 centre[37], as shown in Fig. 2d. The corresponding positions reveal that defects A and B are also present in the photocurrent image. We measure a pulsed PDMR spectrum on defect A by alternately applying a 1-μs-long square laser pulse at 20 mW and an RF pulse with a scanned frequency under a static magnetic field. The two peaks appear at identical frequencies as in the ODMR (Fig. 2e), demonstrating the electrical detection of a single V2 centre. We then characterise the photocurrent generated from the single V2 centre (Fig. 2f). In the previous study[10], the two-photon ionisation mechanism has remained a hypothetical proposal because the observed photocurrent included responses from defects other than the V2 centre. In this study, a quadratic laser power dependence is observed at low laser powers from a single defect, providing experimental evidence of the two-photon ionisation mechanism. At higher laser powers, the dependence transit to linear above the power $P_0$ where the excitation from the

ground state (GS) to the excited state (ES) saturates[40]. This dependence can be expressed as $I(P) = \alpha P^2 / (1 + P/P_0)$, and the fitting suggests $\alpha = 20.1 \pm 2.8$ fA/mW$^2$ and $P_0 = 451 \pm 65$ μW. The linear coefficient of the current–power dependence is $\alpha P_0 = 9.09 \pm 0.07$ fA/mW. These parameters will be analysed later to infer the photoionisation dynamics. We also characterise the photoelectrical imaging spot size of a single V2 centre. Figure 2g compares the laser power dependence of the full-width half maximum (FWHM) of the defect A with optical and photoelectrical imaging obtained by a fitting with a Gaussian function. The photoelectrical imaging spot size is small below $P_0$ due to the two-photon ionisation[14], but increases at higher laser powers. However, in contrast to the monotonically broadening spot size in optical imaging due to the photoluminescence saturation, the spot size of the photoelectrical imaging saturates at higher laser powers. We can model the photocurrent distribution by assuming the spatial distribution of the input laser to be Gaussian and the $I(P)$ relation to be quadratic-linear as discussed above. From this model, we infer the FWHM of the spot in two ways; the first method is the fitting of the calculated photocurrent profile by a Gaussian function, and the second is the analytical FWHM of the modelled photocurrent: $w_{img}(q) = w_{laser} \sqrt{\{\ln[(4q+4)/(q+\sqrt{q^2+8q+8})]\}/\ln 2}$, where $w_{laser}$ is the laser's FWHM and $q = P/P_0$. These two FWHM estimations are also shown in Fig. 2g and agree well with the experiments, which further confirms the validity of the two-photon ionisation mechanism. As expected from the quadratic-linear photocurrent characteristics, the analytical model gives $w_{img}(q \ll 1) = w_{laser}/\sqrt{2}$ and $w_{img}(q \gg 1) = w_{laser}$. Therefore, the photoelectrical imaging resolution is atmost limited by the laser's spot size even at high laser powers where a strong signal can be obtained for imaging with a higher signal-to-noise ratio. We infer the laser's FWHM to be approximately 0.49–0.5 μm.

After confirming the two-photon ionisation mechanism, we can identify the ionisation path. In the previous study[10], it has been unresolved whether the ionisation occurs from the ES or the metastable state (MS) because of the removed photocurrent baseline in the lock-in detected signal due to the RF intensity modulation and the complicated charge transport in the device with many trapping defects. We observe identical contrast signs in ODMR and PDMR of the single V2 centre, as shown in Fig. 2e. This indicates that similar dynamics are involved in ODMR and PDMR. The spin contrast in ODMR is generated by the competitive rate dynamics in the ES between the radiative decay to the GS and the spin-dependent intersystem crossing (ISC) to the MS. In PDMR, the same (opposite) sign in the spin contrast is obtained if the ionisation occurs while the system is populated in the ES (MS), as pointed out in refs. 10,41. Therefore, we identify that ionisation from the ES is the dominant process.

Although we demonstrate the electrical detection of a single V2 centre, the obtained PDMR contrast is ten-fold smaller than that of ODMR obtained with 0.5-mW laser power. Improving the signal contrast and reducing the current noise is crucial for targeted high-fidelity detection. A strategy is increasing the spin-signal current while decreasing the spin-independent current. Here, we control the laser pulse sequences to generate the photocurrent. However, we observe RF frequency-dependent small crosstalk to the detection electronics. To focus on the laser effect in the experiments below, we measure PDMR with a magnetic field sweep, such that the current component from the RF crosstalk remains constant[10].

### Ionisation pulse control for improving single-spin readout efficiency

Considering the confirmed two-step ionisation and the mechanism of spin-dependent photocurrent generation in ES of the V2 centre (Fig. 1a), the spin-dependent component in the photocurrent is contained only in the initial stage of the laser excitation at high power because spin information is initialised after the decay through

MS or randomised by the ionisation. Therefore, collecting photocurrent after the decay through MS or after the second or later ionisation should be avoided to reduce background and noise currents. Furthermore, ionisation induces inefficient spin initialisation. Time gating is utilised in the photodetection, but the amplifier's bandwidth limits the time resolution in the electrical detection. This problem can be overcome by controlling the excitation rather than the detection: splitting the laser pulse into a high-power ionisation pulse and a low-power initialisation pulse. Hrubesch et al.[16] have reported improved PDMR spin contrast for ensemble NV centres. Here, we demonstrate that the excitation modulation can even improve PDMR's spin detection SNR of single defects beyond the ODMR's shot-noise limit.

We first compare a PDMR spectrum obtained with a simple square laser pulse (1 μs, 32.3 mW) and with a two-step pulse (50 ns, 32.3 mW/850 ns, 1.5 mW) shown in Fig. 3a and b, respectively. These pulses are prepared with an acousto-optic modulator (AOM). Despite the decrease in signal with the two-step laser pulse by 40%, the baseline current decreases ten-fold. As the laser intensity fluctuation induces the noise, a decrease in the background current improves the SNR from 4.8 to 8.6. The contribution to the decrease in the current shot noise would also be effective in more ideal measurement systems with less system noise. For short laser durations, the rise/fall time of the laser is also an important factor for efficient ionisation. The rise time of the AOM is 30 ns, longer than the ES lifetime of 6 ns[35], which may lead to inefficient ionisation. Thus, we also measure the PDMR with a pulsed laser with 3-ns rise time, 39-ns width at 34.5 mW followed by 850-ns initialisation laser pulse at 1.5 mW (Fig. 3c). The obtained SNR is 11.4. This improvement is attributed to slightly enhanced signal intensity and suppressed spin-independent current during the power transient time. Also, observing the PDMR signal with the high-power short pulse further confirms that the dominant ionisation path is from the ES. With the short-pulsed laser, we can read out coherent spin manipulations. We observe a photocurrent oscillation against the RF pulse length (Fig. 3d) and a linear increase in the oscillation frequency with the square root of the RF power (Fig. 3e). The results indicate that the observed oscillation is a Rabi oscillation of a single V2 centre. We find the Rabi oscillation frequency by PDMR and ODMR are in good agreement (Supplementary Note 3).

We also performed Ramsey interferometry and Hahn echo by PDMR on defect A. The Ramsey measurement with the RF detuning of 6.33 MHz (Fig. 3f) exhibits a complex oscillation with three frequency components. A simulated Ramsey signal assuming coupling to two $^{29}$Si nuclear spins ($I = 1/2$) in the fourth nearest neighbour of the silicon vacancy site using the hyperfine parameter reported in ref. 42 ($A_{zz} = -2.20$ MHz, $A_{xx} = A_{yy} = -2.6$ MHz) agrees well with the experiment, and the fitting finds the detuning of $6.296 \pm 0.012$ MHz and the inhomogeneous decay time of $T_2^* = 435 \pm 12$ ns. In Hahn echo (Fig. 3g), we apply first π/2 and π pulses with a fixed interval of $\tau_1 = 1$ μs, and we vary the interval $\tau_2$ between the π and the second π/2 pulses. The appearance of the signal at $\tau_2 = 999.6 \pm 4.0$ ns ($\tau_1 = \tau_2 \gg T_2^*/2$) confirms the detection of the spin echo, and the oscillating signal at $2.186 \pm 0.040$ MHz with a decay time of $449 \pm 21$ ns further convinces the existence of the coupled nuclear spins. As seen in these measurements, PDMR is capable of coherent detection at single-spin sensitivity.

## Discussion

We compare the efficiency of optical and electrical detection. The previous study on a single NV centre reported better SNR in electrical imaging[13], which we find for the V2 centre as well. To estimate SNR in imaging, we measure the photon counts and photocurrent at two points: at the centre and the background of defect A. The signal is the difference of the mean at two points, and the noise is the standard deviation of the data at the background. The photon counts are

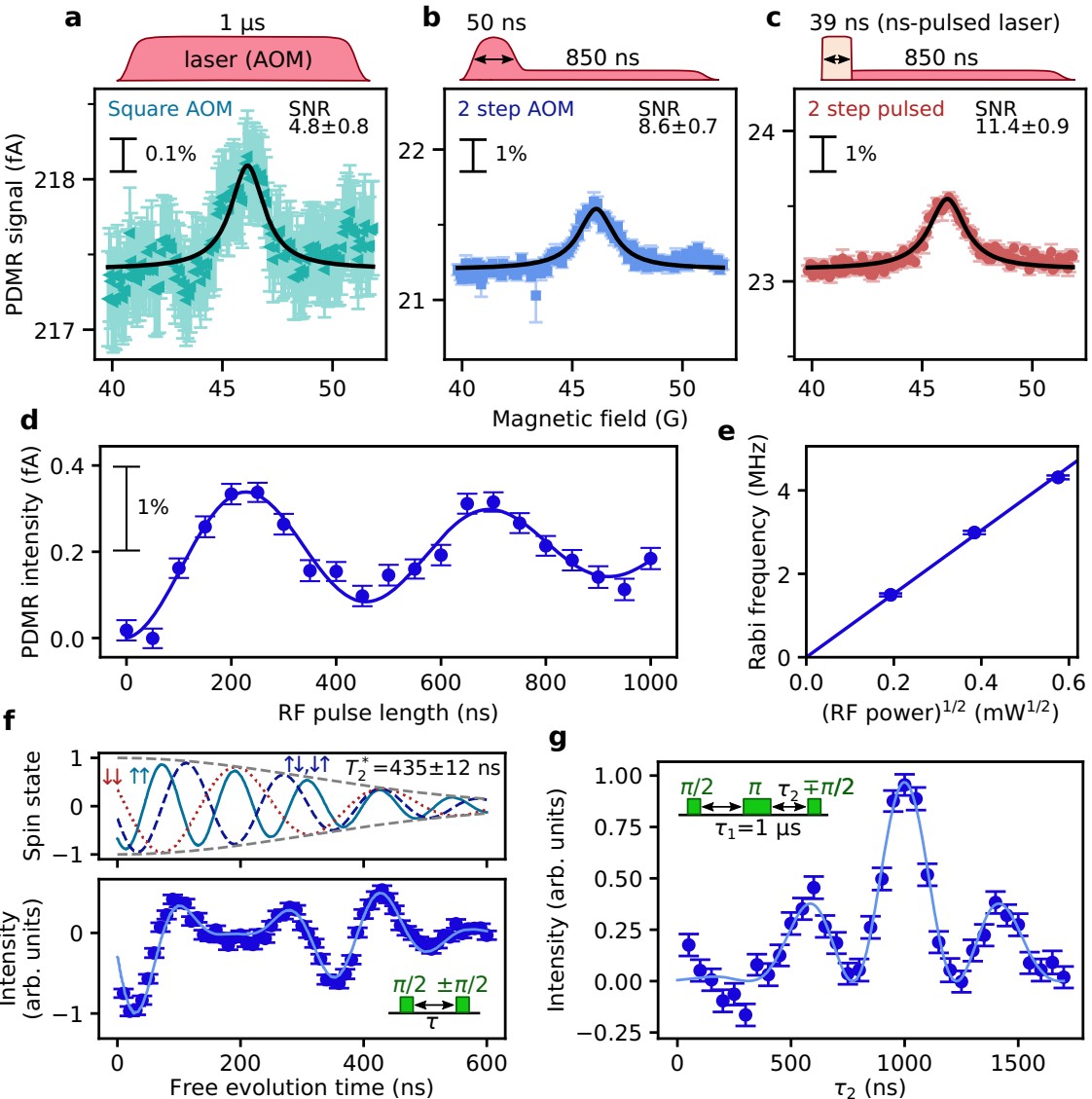

**Fig. 3 | SNR improvement in PDMR with laser-pulse shaping and coherent spin readout.** PDMR spectra of defect A measured with laser pulses: **a** 1 μs square pulse shaped by AOM at 32 mW, **b** 50 ns pulse at 32.3 mW followed by 850 ns initialisation pulse shaped by AOM at 1.5 mW, and **c** 39 ns pulse with 3 ns rise time by ns-pulsed laser at 34.5 mW followed by 850 ns pulse shaped by AOM at 1.5 mW. The experimental data are shown with the raw baseline current and fitted using a Lorentzian function (black lines). The integration time is 120 s for each point. RF frequency is fixed at 199 MHz, and the magnetic field is swept. PDMR sequence is prepared with $T_{wait} = 1000$ ns and $T_{seq} = 2530$ ns (Fig. 2e) for **a**–**c**. **d** PDMR-detected Rabi oscillation of defect A measured with the laser conditions used in **c**. From experimental data, a nonlinear background induced by RF is subtracted. (Methods). The blue line is the fit with a damped cosine function. **e** The RF power dependence of the Rabi frequency. The blue line is a linear fit. **f** PDMR-detected Ramsey interferometry. The dots in the lower panel are the normalised

experimental result measured at RF frequency detuned by +6.33 MHz, and the solid line is the simulation assuming two $^{29}$Si nuclear spins in the fourth neighbour shell. The upper panel shows the simulated electronic spin state decomposed by the nuclear spin components. The inhomogeneous decay is assumed to be $\exp[-(\tau/T_2^*)^2]$, and $T_2^*$ is estimated by fitting the simulation to the experimental data. A linear background in the experimental data is corrected based on the simulation result. **g** PDMR-detected spin echo. The time interval $\tau_1$ between the first RF π/2 pulse and the π pulse is fixed at 1 μs, and the interval $\tau_2$ between the π pulse and the second π/2 pulse is varied to observe the echo signal. The dots are the normalised experimental data with a correction of the linear background. The solid line is the fitting curve with a function $\cos^2[\pi f(\tau_2 - \tau_0)] \exp\{-[(\tau_2 - \tau_0)/T_2^*]^2\}$, where $\tau_0$ is the timing of the echo refocus. All error bars in this figure show ±1 standard error.

measured with the integration time bin width of 62.5 ms, a close value used in Fig. 2a and d, and the photocurrent data is sampled at the same time interval from the lock-in amplifier. Figure 4a shows the SNR comparison in optical and electrical imaging, showing non-saturating electrical SNR at increased laser power exceeding the optical SNR, which is limited by the fluorescence saturation and the increasing background. Here, we focus on the SNR in spin detection, which is important for quantum technologies. To perform the comparison, we first optimise the ODMR method as discussed in Supplementary Note 3, and we achieve a photon shot-noise-limited SNR. We compare

the SNR of ODMR and PDMR in Fig. 4b and c for defect A. In addition, we measure defect C with slightly larger zero-field splitting than V2, probably a modified silicon vacancy[43] (Supplementary Note 5). PDMR exhibits 1.7 and 2.0 times higher SNR for defects A and C than ODMR, respectively. The SNR of ODMR can be improved by a factor of 1.2 if the background fluorescence is removed, but further improvement requires improved photon collection. The 1.7–2.0 fold improvement in SNR corresponds to a 3–4 times more efficient photon collection. The photon collection enhancement is achievable with photonic structures, such as a solid immersion lens (SIL)[18,44–46], a nanopillar[47–50], and

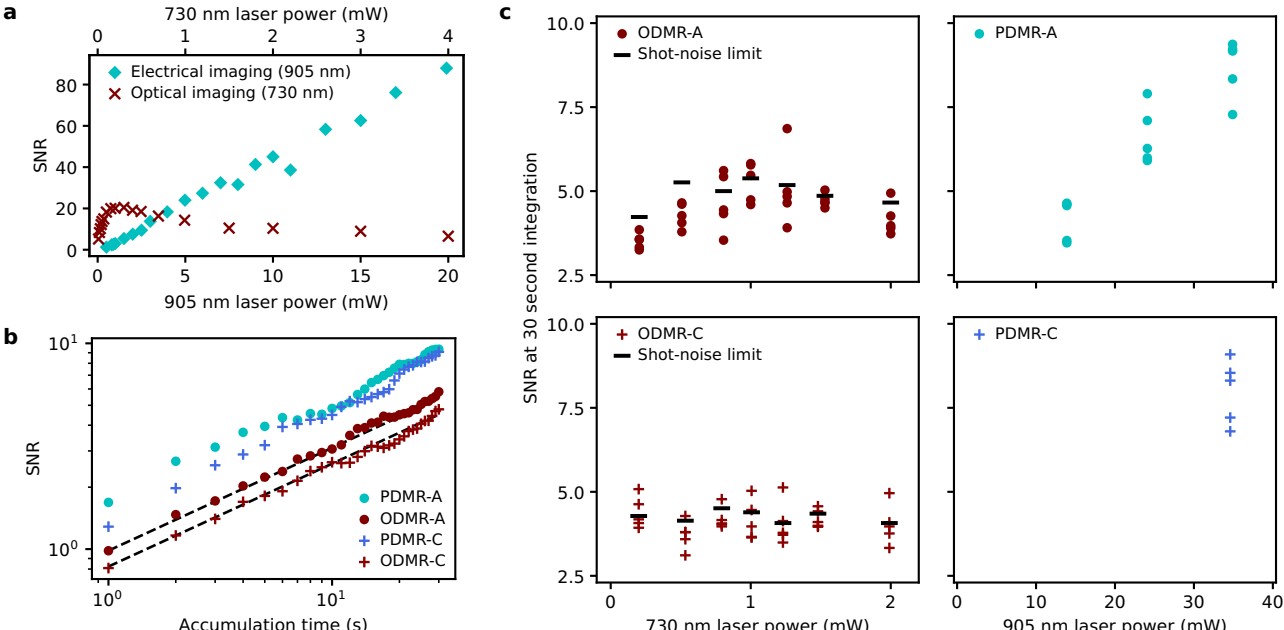

**Fig. 4 | SNR comparison between electrical and optical detection. a** Laser-power-dependent SNRs in imaging. **b** Spin-readout SNRs in PDMR and ODMR measured at two single defects, A (V2 centre) and C (V2-like defect). Each SNR is estimated from the magnetic-field-sweep spectrum. The same RF conditions, wait time after the laser pulse, and the total sequence length as in Fig. 2a-c are used. For ODMR, the laser power and photon counting window are optimised to obtain the best SNR (Supplementary Note 3). Black dashed lines are photon-shot-noise-limited SNRs in ODMR for each defect. For PDMR, the fast 39 ns pulsed laser is used for ionisation followed by an 850 ns initialisation laser pulse at 1.5 mW. The ionisation laser power was 34.9 and 34.6 mW for defects A and C, respectively. **c** Laser-power-dependent SNRs of ODMR and PDMR with the two-stepped pulse (various-powers 39-ns-width pulsed laser/1.5-mW 850-ns-width pulse) of defects A and C. The data are accumulated for 30 seconds per data point, and five measurements are performed per laser power condition. Black bars are the shot-noise-limited ODMR SNRs corresponding to accumulation for 30 seconds calculated from the data accumulated for 150 seconds per data point.

an optical waveguide[51–54], but they require intensive fabrication. The SNR improvement with PDMR is comparable to the enhanced photon collection in SILs for single silicon vacancies[18,44]. However, PDMR only requires electrode fabrication, which is advantageous in device integration and miniaturisation.

The SNR of PDMR can be improved further in two aspects. One aspect is reducing technical noises. We observe a noise of 0.41 fA/Hz$^{1/2}$ in the PDMR baseline measured with a short pulse laser, which is 4.3 times larger than the photocurrent shot noise. Our system noise is 1.7 times larger than the characterised amplifier noise of 0.24 fA/Hz$^{1/2}$. We find that the major noise source is the laser, measured to be 0.24 fA/Hz$^{1/2}$ by blocking the laser illumination. The remaining noise is from the electronic setup for controlling the experiments 0.20 fA/Hz$^{1/2}$. The RF-induced random noise is estimated to be 0.1 fA/Hz$^{1/2}$. Therefore, stabilising the laser intensity and eliminating the remaining technical noises would improve SNR up to the amplifier and shot-noise limit. The other aspect is increasing the photocurrent, which we discuss below. The signal detection efficiency is a product of the ionisation and carrier collection efficiencies. As we estimate the latter to be approximately unity based on the saturated photocurrent–voltage characteristics (Supplementary Note 1), the ionisation efficiency currently limits the detected charges. In PDMR, one photoionisation cycle generates a current corresponding to one elementary charge. As the sequence repetition rate is 395.3 kHz in the PDMR experiment, if one laser pulse triggers one photoionisation cycle, the fundamental frequency component of the electrical current detectable with the lock-in amplifier is 28.5 fA (to convert from a square waveform amplitude to lock-in detection amplitude, a factor $\sqrt{2}/\pi$ is considered). Assuming ≈5% PDMR contrast (2.0% in Fig. 2c and corrected for the background and initialisation currents), we therefore expect 1.4 fA spin signal intensity. Compared with the experimental intensity (0.51 fA), the estimated ionisation or collection efficiency is roughly 40% given the current

conditions. As the spin signal intensity does not saturate at maximum laser power in this study (Supplementary Fig. 6d), the detected signal is limited by the ionisation efficiency. This value is already higher than a typical photon collection in confocal microscopy (a few percent)[15], but there is room for further improvement by increasing the laser power. Indeed, the SNR of PDMR is not saturated at increased laser powers, as shown in Fig. 4c, whereas the SNR of ODMR shows a peak at around 1 mW. To understand the achievable SNR based on the charge dynamics, we estimate the ionisation rate and simulate PDMR dynamics using the obtained rates and a simplified model (Supplementary Note 6 and 7, respectively). In this analysis, we use a four-level model considered for the NV centre in diamond[40], including the GS, ES, and MS of the V2 centre and the neutral state as depicted in Fig. 1a. We do not distinguish the spin states in the GS and ES for the simplicity. In this model, the steady-state photocurrent $I_{ph}$ is described as $I_{ph} = e\eta_c n_{ES}\gamma_i P$, where $\eta_c$ is the charge collection efficiency, $n_{ES}$ is the population of the ES, $\gamma_i$ is the ionisation rate per unit laser power, and $P$ is the laser power. As the laser power increases, $n_{ES}$ saturates as $n_{ES} \propto P/(1+P/P_0)$ due to the shelving into the MS and the ionisation-recharging dynamics, and therefore $P_0$ and the coefficient $\alpha$ discussed in Fig. 2f can be described by the rates (for details, see Supplementary Note 6). Therefore, we can estimate both $\gamma_e$ and $\gamma_i$ from the laser power dependence of the photocurrent as shown in Fig. 2f. By solving the steady-state rate equation for the four-level model, $\gamma_e$ and $\gamma_i$ are written as $\gamma_e \approx \{\tau_{ES}P_0(1+k_2/k_3)\}^{-1}$ and $\gamma_i = (\pi\alpha P_0/\sqrt{2}e\eta_c)(1+k_2/k_3)$, respectively, where $\tau_{ES} = (k_1+k_2)^{-1}$ is the ES lifetime, and $k_1$, $k_2$, and $k_3$ are decay rates from the ES to the GS, the ES to the MS, and the MS to the GS, respectively (Supplementary Note 6). Here, we use the approximation that the recharging rate from the neutral state to the GS is much faster than $\gamma_i$ based on the photostability of single V2 centres[18]. By using the reported spontaneous decay rates[38,55] and the measured $\alpha$ and $P_0$, we calculate $\gamma_e^{-1} = 17 - 24$ ns · mW and $\gamma_i^{-1} = 0.9 - 1.1$ µs · mW at

905 nm under the assumption of $\eta_c \approx 1$. Using the excitation and ionisation rates calculated above and the spin-dependent internal transition rates in ref. 56, we simulate the photocurrent and the PDMR signal intensities for the experiments of pulsed PDMR with the nanosecond short laser pulses by solving the rate model. Although this calculation is independent of the estimation of the ionisation efficiency described above, our simulation also indicates a photoionisation efficiency of around 40%. This suggests that the assumption of high charge collection efficiency close to unity is reasonable. Further, the simulation suggests further possible improvement in SNR by a factor of three if a higher ionisation rate is achieved. As the ionisation cross-section at 905 nm is found to be about 50 times smaller than the absorption cross-section from GS to ES at 905 nm, the PDMR could be further improved by searching for the optimum wavelength for efficient ionisation[57].

Finally, we briefly mention the additional isolated single-defect-like photocurrent features that do not appear in the optical image. Spot X shown in Fig. 2d, which only appears in the photocurrent measurement, has a comparable size and intensity to those of defects A and B (Supplementary Note 4) without a clear PDMR at V2's resonance. Considering the excitation wavelength (905 nm), X can be another single defect with (near-)infrared photoluminescence out of the silicon detectors' spectral range (above 1060 nm), such as divacancies (zero phonon line: 1078–1132 nm)[58] or nitrogen-vacancy centres (zero phonon line: 1131–1242 nm)[59]. Another possibility is an optically inactive centre. Although PDMR in SiC has been demonstrated only for silicon vacancies so far, these results suggest that the electrical detection may provide individual defect sensitivity for other interesting defects. Also, it suggests the possibility of searching for optically unidentified defects by PDMR using various laser wavelengths.

In summary, we realise the electrical coherent readout of an electron spin of a single V2 centre in 4H-SiC at room temperature. We demonstrate a higher SNR in the electrical detection than in the optical method. The SNR enhancement by PDMR is comparable to ×3–4 photon collection improvement in ODMR. In addition, optimising the laser pulse and peripheral circuit would further improve SNR, potentially exceeding optical detection with a photonic structure. Such PDMR's potential for highly sensitive readout is crucial to realise the high-fidelity qubit readout for many quantum applications. The demonstrated single spin detection in PDMR in SiC is of significant importance for enabling integrated and miniaturised quantum information and high spatial resolution quantum sensing devices. Also, the observed single-defect-like photoelectrical features without photoluminescence in the silicon vacancy's emission spectral range indicate the possibility of applying PDMR for other types of individual (near-)infrared defects and exploring optically unidentified systems. These prospects apply not only to defects in SiC[60,61] but also to other colour centre systems in various wide-bandgap materials such as diamond and gallium nitride[62].

## Methods
### Sample preparation
The sample is an epitaxially grown $p$-type 4H-SiC. The epitaxial growth is conducted using a vertical hot-wall CVD reactor with a commercial 8° off (0001) substrate[63,64]. The gas system is $H_2$-HCl-$SiH_4$-$C_3H_8$, and the system pressure is maintained at 27 kPa. The substrate temperature is held at 1873 K during the epitaxial growth. The flow rates of $H_2$ and $SiH_4$ are 40 and 0.1 slm, respectively. The C/Si and Cl/Si ratios are set at 1.1 and 10, respectively. The net acceptor (presumably boron) concentration is determined to be $2 \times 10^{14}$ cm$^{-3}$ by capacitance-voltage characteristics.

The single silicon vacancies are created by electron-beam irradiation with the fluence of $8 \times 10^{13}$ cm$^{-2}$ at 2 MeV and post-annealing process at 600 °C for 30 minutes in Ar. After annealing, the sample is cleaned with a piranha solution. Although silicon vacancies are created throughout the epitaxial layer at 2 MeV, negatively charged single silicon vacancies are found from the surface to a depth of about 3 μm, probably due to the surface band bending in the p-type material. A metal-semiconductor-metal diode device is fabricated for PDMR. We designed the electrode gap to be about 7.5 μm to have saturated charge collection efficiency at around 10 V. We first deposit the 200-nm-thick $SiO_2$ film by plasma-enhanced chemical vapour deposition with tetraethyl orthosilicate precursor. Then, the RF antenna and bonding pads (Au/Cr = 150 nm/10 nm) are fabricated by photolithography and lift-off on $SiO_2$ to isolate them from SiC. Here, Cr is inserted as an adhesive layer for stable wire bonding. The SiC surface is exposed for active areas of the devices by photolithography and wet etching with buffered HF solution. After removing the photoresist with organic solvents, the pattern of the contact electrodes is defined by photolithography, and then 100-nm-thick Au electrodes are deposited by electron-beam evaporation, which is followed by the lift-off process to form a pair of Schottky diodes. The sample is not annealed after the Schottky contact formation. We observe the saturation of the detected current at 10 V for the device with a 7.1 μm gap, showing a good agreement with the target voltage (Supplementary Note 1). The fabricated sample is mounted and wire-bonded onto an in-house fabricated printed circuit board with a microstrip line structure for RF, with its ground plane electrically connected to both the RF line ground and the signal line ground.

### Measurement setup
The experiments are performed with a home-built PDMR spectrometer based on a confocal laser scanning microscope. For the optical measurements, the 730 nm excitation laser illuminates the sample via an oil-immersion objective lens (Nikon Plan Apo lambda 100x, NA = 1.45) fixed on a three-axis piezo stage. The laser can be modulated with an acousto-optic modulator (AOM) for pulsed ODMR measurements with a rise time of 12.5 ns (10–90%). The fluorescence is collected through a long-pass dichroic mirror (Thorlabs DMLP805), 100 μm diameter pinhole, and long-pass filters of 830 nm and 900 nm cut-off wavelengths to block the laser breakthrough and Raman scattered light. Afterward, the fluorescence is split into two paths by a polarised beam splitter (PBS) and separately detected by two single-photon-counting avalanche photodiodes (Excelitas SPCM-AQRH-14 and -16, APDs) at the end, which constructs a Hanbury Brown-Twiss interferometer to identify single silicon vacancies. For the autocorrelation measurements, Glan-Thompson prisms are placed between the APDs and the PBS to suppress the crosstalk of the unpolarised breakdown flash from the APDs. The photon number detected by APDs is counted by a time tagger (Swabian instruments).

For photoelectrical measurements, two 905 nm lasers are used: a continuous-wave laser amplitude modulated by an AOM and a nanosecond pulsed laser. The rise times (10–90%) of the AOM and the nanosecond pulsed laser are characterized as 30.1 and 3.3 ns, respectively, with a photodiode and an oscilloscope with 1 GHz bandwidth. The lasers are coupled into a single-mode fibre and then directed to the sample via the same objective lens as optical measurements. The excitation paths for optical and photoelectrical measurements are switched by a flip mirror located before the objective lens. A bias voltage is applied between the electrodes, and the photocurrent is amplified by a $10^{12}$ V/A transimpedance amplifier (NF SA-609F2) and then lowpass filtered with a 100 Hz cut-off frequency (Stanford Research Systems SR560). Finally, the signal is detected by a lock-in amplifier (Stanford Research Systems SR830). In the photoelectrical imaging, the continuous wave laser is modulated into a square wave at 13 Hz for lock-in detection.

The radio frequency (RF) to manipulate the electron spin state is pulsed by an RF switch and delivered to the V2 centre via the fabricated

antenna after amplification. The external magnetic field is applied by an electromagnet. The field direction is aligned along the c-axis of the sample within 3 degrees. The calibration of the electromagnet is performed by ODMR measurements of the resonance frequency of the V2 centre. The static magnetic field for Figs. 2e, 3d, and e is 45.8 G, and that for Fig. 3f and g is 57.9 G. The swept magnetic field range for Fig. 3a–c and Fig. 4's data is from 39.2 G to 51.5 G.

The laser pulse sequences and RF pulse timing for ODMR and PDMR are generated by an arbitrary waveform generator (Tektronix AWG5014C). For PDMR experiments, the bandwidth of the system is limited by the transimpedance amplifier. Therefore, we repeat short pulse sequences and modulate the laser amplitude with a slow square envelope at 7 Hz for lock-in detection[65].

The transmittance of the objective lens at 730 nm and 905 nm is 81% and 74%, respectively. All the laser powers written in the article are the values measured before the objective lens and not corrected for transmittance.

### Background subtraction of Rabi measurements

In the PDMR-Rabi measurements, we observe the RF-pulse-length- and RF-power-dependent drift in the background. To subtract these backgrounds and obtain the clear Rabi signals, we conduct alternate measurements at on-resonant and off-resonant fields and then subtract them. The difference in the magnetic fields between on- and off-resonance conditions corresponds to approximately 9 MHz in the ODMR peak frequencies.

### Ramsey and Hahn echo measurements

PDMR Ramsey and Hahn echo are performed by using an ns-pulsed ionisation laser pulse and a 600 ns long spin initialisation laser at 0.5 mW. The $\pi/2$ and $\pi$ pulse lengths are 39 and 77 ns, respectively. The wait time between the laser and the RF sequence is 500 ns. To cancel out the common-mode noise[12,41], two PDMR signals with the inverted RF phase of the final $\pi/2$ pulse are measured $S(\pm)$. $S(+) - S(-)$ and $S(-) - S(+)$ are plotted in Fig. 3f (Ramsey) and 3g (Hahn echo), respectively. The simulation of Ramsey measurement is performed using QuTiP[66,67].

## Data availability

The data of this study have been deposited in the Kyoto University Research Information repository https://doi.org/10.57723/kds587378.

## Code availability

The codes to fit the experimental data and for numerical simulation are available from the corresponding authors upon request.

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

## Acknowledgements

We thank Ernst David Herbschleb and Masanori Fujiwara for the useful discussions and valuable insights. This work was supported by MEXT Q-LEAP Grant Number JPMXS0118067395 (N.Mi and T.O.), JSPS KAKENHI Grant Numbers JP23K22796 (N.Mo), JP21K20502 (N.Mo), and JP23K19120 (T.N.), JST PRESTO Grant Number JPMJPR245C (N.Mo), JST SPRING Grant Number JPMJSP2110 (T.N.), research grant from Murata Science and Education Foundation (N.Mo), the Spintronics Research Network of Japan (N.Mi), and the Future Development Funding Programme of Kyoto University Research Coordination Alliance (N.Mo). This work was supported by Kyoto University Nanotechnology Hub in "Advanced Research Infrastructure for Materials and Nanotechnology Project" sponsored by MEXT, Japan, for device fabrication.

## Author contributions

N.Mo and N.Mi conceived the project. K.M. and H.T. prepared the SiC sample. H.A. and T.O. performed the electron-beam irradiation. T.N. and N.Mo designed and fabricated the device. N.Mo designed and programmed the measurement system. T.N., N.Mo, and K.O. constructed the setup. T.N. and N.Mo performed the measurement and the data analysis. N.Mi supervised the study. N.Mo conducted the numerical simulation. T.N. and N.Mo discussed the result with N.Mi and wrote the manuscript with contributions from all authors.

## Competing interests

The authors declare no competing interests.
