## [Transparent Peer Review file · Nature Communications]

Coherent photoelectrical readout of single spins in silicon carbide at room temperature

Corresponding Author: Professor Naoya Morioka

Version 0:

Reviewer comments:

Reviewer #1

(Remarks to the Author)

This work demonstrates the photoelectrically detected magnetic resonance (PDMR) measurement of a single silicon vacancy spin in 4H-SiC, wherein the magnetic resonance of a single spin can also be optically detected, known as optically detected magnetic resonance (ODMR). By optimizing the ionization pulse control, the signal-to-noise ratio (SNR) of PDMR is shown to be 1.7-2 times higher than that of ODMR. The underlying mechanism is analyzed based on the transition processes of silicon vacancy electrons under excitation. The demonstrated electrical single spin qubit readout method in integrated semiconductor materials could further extend the application field of spin defects in SiC. However, there are several points that the authors need to clarify before I can recommend publication in Nature Communications:

1. The sample is a p-type 4H-SiC with a dopant density of $2 \times 10^{14} \text{ cm}^{-3}$. Is the dopant density related to the electrical readout efficiency? Would similar results be obtained for an N-doped sample?
2. Although no signal was detected when measuring the ODMR spectrum of spot B, is it possible that the spin of spot B could be read out using the PDMR method? If so, this would represent a significant advantage of PDMR over ODMR.
3. It would be valuable to measure the Rabi oscillation using both the ODMR and PDMR readout methods under the same microwave power and compare the Rabi oscillation frequencies obtained with these two readout techniques.
4. For the Rabi oscillation shown in Fig. 3(d), it would be appropriate to include a scale bar representing the signal contrast, similar to those in Figs. 3(a), (b), and (c).

Reviewer #2

(Remarks to the Author)

The authors present an interesting work on photo-electrical readout of a single V2 centre spin state in SiC with a potential of using it as a spin qubit. The work appends on previously successfully established photo-electrical readout of V2 ensembles (Ref. 10) in SiC and using the technique developed in Ref. 9 for NV diamond. The successful demonstration of single V2 centre readout is possible to achieve due to the optimisation of the experiments, and the spin initialisation and readout by tailored laser pulses, adopting and improving the technique developed in Refs. 16 and 9. However, although this work is highly relevant for development of the solid state quantum technology and for miniaturisation of quantum devices, it is based on improvements rather than coming with novel break-through ideas. Also there are some very basic misunderstandings in semiconductor physics, in otherwise excellently carried out quantum optics experiments, in particular see below.

i. One of the novelty claim is to use the Schottky contacts, to avoid to employ what the authors call " Ohmic annealing", but this claim is not significant. For example Schottky contacts have been used in ref. 16 for NV centres, but which type of the contacts is best to be used is just a technical decision to be made for concrete experiment as both (injecting, Schottky) are very standard device configurations. On opposite ohmic contact would lead to the photocurrent gain, so signal can be much higher than for Schottky contacts, so it is matter of choice and it is mainly important how the ionisation/recombination and the charge carrier transport are used in the device. Also annealing is used in the special case of diamond to engineer ohmic contacts by forming a charge carrier tunnelling barrier through 10 nm thin TiC, produced from Ti during the annealing. However even without annealing, TiC can be formed on diamond but the layer is thinner. So annealing is not a must and avoiding it is no significant idea which would make the paper more impacting. SiC is different material than diamond and with significantly lower gap so ohmic contact forming is easier but even though for some Schottky contacts on

SiC the annealing is necessary. But these are technical aspect which can be used in Supplementary. Also the contact formation using Au is poorly documented.

ii. There are significant questions concerning device physics. The authors state that they operate Schottky diode and reach the flat-band conditions at 9.1V, for which they design contact geometry, taking into the account the background acceptor and hole concentrations. They also present in SI IV characteristics to support this statement. However, the situation is much more complex and basics devices physics, to which the authors refer, is not well interpreted. Referring to Fig 1, the authors have no single Schottky diode but two of them back to back connected, i.e. one of two (in dark) is always blocking. If the flat band conditions occur at some voltage V_b (equal to barrier height) on one Schottky diode, the potential on the other diode is equal to two barrier heights. So one can not speak globally about the flat band conditions in this situation where there is no uniform potential across the sample throughout the gap between the contacts (the second contact has a double height barrier). Secondly, the authors misinterpret the term of flat-band conditions and potential for the full depletion (which are two very different terms). Flat band conditions, as stated before, occur if the applied voltage in forward direction is equal to the barrier height. As the barrier height for Au on SiC is about 1.5 V. The flat band potential must be the same (can be slightly influenced by the interface state charging). So flat band conditions can not simply be 9.1 V. the authors misinterpret this term in place of stating depletion. Also for voltages above the flat band the photocurrent saturation cannot be achieved as the contact becomes injecting, but the saturation is due to the full collection and to the fact that back to back (ie always in reverse) Schottky diodes are used. Actually the barrier would becomes injecting i.e.will be ideally Ohmic in flat band conditions. These are quit basics semiconductor physics terms which need to be put right.

also, the IV characteristics in the supplementary is measured only in one direction and should be measured in both directions to diagnose in which model the device actually operates. As mentioned, the fact that the author see the photocurrent saturation means

that the device operates in non-injecting, i.e no flat-band condition. However the claim that the collection efficiency is one (which can not again be in the flat band conditions), taken from basics semiconductor book is not convincing. This assumes no recombination. To claim such high collection efficiency, the authors need to precisely characterise the junction, (measure both barriers) show the mentioned CV plots (but they are not presented and should be both in dark and under illumination using different laser power) to provide full device characteristics. The IV characterises in Si do not point to such high collection efficiency (and should be fitted with the diode factor).

Also the authors might forgotten in their description about the fact that both electrons and holes i.g. a charger carrier pair is generated from V_2 . This means that one type of the photo carrier can in reality can cross one of the barriers, while the second polarity of the charge the second barrier. This should be implemented in the models. This could bring to my opinion bring the paper needed innovation and allow to discuss the full photoionisation and transport cycle in the benefit of justification of the publication in Nature Comm.

lii. The authors mention that one photoionisation cycle generates current of one elementary charge. This is not precise it is one electron-hole pair which is generated. However the discussion of the photoionisation efficiency is unclear. The author state that ionisation cross section is about 50 times smaller than the absorption cross section. Which absorption the authors exactly mean, the ZPL absorption? Finally if the second step, i.i. photoionization is 50 times smaller than the first step, it means that 50 x more photons are needed for this step. How then the current collection can be (with respect of incoming photon numbers) 40% ? In the full rate equation model it means, that due to this 1:50 ratio, only 1 electron of 50 is used for the photoionisation. This can be in general expressed in the term of QE for PL and QE for photocurrent which have to be compared (as a function of the laser power). These numbers and consequently the theoretical S/N ratio do not match. The authors need to express the number of generated terms in the framework of QE.

This situation is different to diamond where the QE for PI is about 80%, While QE for photocurrent is about 50%. For the case of V_2 in SiC, based on the author rate comparison it seems to be much lower, and authors need to clarify it.

iv. There are still some other technical issues. The methodology for the contact preparation needs to be described as the surface plays in this process an important role and contacts need to be characterised. Also in Supplementary the photo I-V is sows the signal less than the dark current, is the dark current from the signal case subtracted, please clarify.

Further on, there are a few important details concerning the detection. The authors use the amplifier with the amplification $10E11$, for this the bandwidth is typically close to 1 second, the authors should clarify/put reference to methodology how they can measure so fast signal after the illumination with laser pulses on 40 nanosec (or 1000 nanosec) range and Rabi period in the range of 100 nanosec. Also in their pulse sequence the RF is applied after 1000 nanosec after the laser pulse. what is the coherence time of V_2 in that case? These are very important parameters to list if one discusses the potential quantum technology applications.

From the comments above, the paper is by far not ready for any publication (and has basic device physics errors). However I estimate that it can be improved by the full rate modelling using the electron-hole pair generation, drift and the collection by the electrodes and back to back Schottky description . Also the improvement in SNR compare to ODMR is minor (it can be improve by a SIL by a factor 50) and the authors should discuss the levels to which it could be enhanced for example in diamond this ration is about 1000). Also it should be discussed how was one can detect the signal to achieve SQL as for photon counting which limits, jointly wioth a extremely low current detected, real application for single SiC V_2 qubits, so limits and improvements need to be clearly discussed. Die to deficiencies in device physics, for the moment, the paper can not be published elsewhere. Also the authors use a model from Ref 10, but this is a tentative model, so rate modelling can be off scale.

Reviewer #3

(Remarks to the Author)

Nishikawa et al report photoelectrical detection and coherent control of single V- centres in 4H-SiC. This is a first, as to date photoelectric detection has been applied only to ensembles of defects, though was done for single NVs in diamond some time ago. The paper also examines improvements in signal to noise attainable with photoelectrical detection, and shows evidence of better SNR using PDMR than ODMR by using short, high power pulsed light. The paper reports interesting results that I think will be of interest to the community, but I can't find a compelling case for publication in Nat Comms just yet, and so cannot recommend publication at this time. My concerns are listed below, and if satisfactorily addressed the paper could be acceptable.

1. PDMR in SiC was first demonstrated in 2019, and single-NV photoelectric readout in the same year. While the authors report the first photoelectric detection of single centres in SiC, it is not clear what particular challenge or roadblock was overcome and why PE detection of single centres is now possible: the experiment seems very similar to the previous work (including by the same group, ref 12). Is this related to near-surface charge stabilisation of single defects, as mentioned in the supplementary information? If so this point should be made clearer, or any other reason why single defect PDMR is so much harder in SiC.
2. Much emphasis is put on the improvements possible with PE detection and advantages over conventional optical detection. The supposedly enhanced SNR (something like 60x for PE compared to optical according to fig 4) is not at all evident in Fig 2a,d, I would say the confocal optical measurement is far higher SNR. Given the P^2 two photon ionisation dependence, the PE detection features should be smaller than those in standard PL. While the signal to noise is well defined for the spin measurements, how is noise defined in the imaging measurements - is background photocurrent the noise? Why is PDMR in Fig 2 so much better than in Fig 3, when the higher SNR scheme is presumably used in the latter?
3. I think the enhanced SNR the authors mention so much would need to be put to the test in a more useful setting than just PDMR. I don't think Rabi alone quite cuts it for demonstrating coherent control, and if SNR can be enhanced so much why are measurements like Ramsey and Hahn echo not done? Also, the authors say on lines 148-149: "Despite that the signal decreases with the two-step laser pulse by 40%, the baseline current decreases ten-fold": I don't see that in the data, has it been subtracted away?
4. The discussion and conclusion sections appear to state that electrical detection is somehow simpler than optical measurements: "PDMR does not require extensive fabrication", "efficient detection with PDMR requires only electrode fabrication" - I really don't buy this, electrical detection is far more of an involved process than optical detection and this paper doesn't convince me otherwise. I think the authors should instead highlight the outlook their technique offers for integrated devices, and why single defects represent a significant advance.
5. The authors mention being limited by unknown electrical noise to almost twice their amplifier noise level, but do not suggest the possible sources of this noise or outline mitigation strategies: is this rf rectification related, and that is why the B field is swept rather than the rf freq?
6. The results of the simulations detailed in the Supplement are quite tersely reported in the main text and difficult to follow (lines 208-212): did the simulation predict a 40% ionisation efficiency or did it use this result from the experiment? Is the suggested procedure to "enhance the ionisation" equivalent to finding a better wavelength, as is mentioned soon after? The authors might also consider citing Todenhagen and Brandt arXiv:2307.11830 here. I think the paper could also benefit from moving some of the details in the supplement into the main text.
7. The authors mention a preference for Schottky-type contacts in contrast to the oft reported need for ohmic contacts. The reference to the flat-band condition is particularly interesting, and this point alone is of great interest to anyone working in PDMR particularly in materials such as diamond. I would like to see a few extra details included in the paper (or perhaps a more thorough dataset in the Supplement) that explain how 100% charge collection efficiency is obtained or even possible: in the supplement it is suggested that this arises from the carrier lifetime only, however it is now well demonstrated in diamond that charges emitted from single defects can be captured by other defects, eg. Lozovoi et al Nature Electronics 4 (10), 717-724 2021. Other work has explicitly considered the role of charge capture of luminescent and nonluminescent defects in photocurrent measurements, eg. arXiv:2402.07091. In SiC at room temp these effects are convolved with a significant acceptor and free carrier concentration in contrast to diamond, so I am curious as to just how many charges generated by a photoionisation cycle end up in the detection electrodes.
8. The authors mention that under 730nm illumination, no isolated single site photocurrent sources are identified "probably due to a large background from the interband two-photon electron-hole pair generation". While the powers used are high (30mW+), are they really high enough to do two-photon band-to-band excitation, even when in ns pulses and confocal volumes? Are the authors able to provide a reference or formula for this?
9. I'm curious as to the identity of the mystery defect-like feature near the V2 centre. The authors mention possible candidates such as an N-V defect or divacancy, but should also mention the emission characteristics of these centres as well, i.e. ZPL, PSB wavelengths.

10. The lock-in frequency is very low for PDMR - 7Hz in Fig 2 and 13Hz in the methods. The amplifier has bandwidth up to 300Hz at 10^{-12} V/A, could the authors explain their choice of such a low lock in frequency?

Other minor points:

1. The authors refer to fluorescent features in the PL and PE detection images as "spots" a few times out of context (eg in the abstract, "We also observe photocurrent spots without photoluminescence in the spectrum range of silicon vacancies", on line 112 "We successfully obtain a clean spot-like photoelectrical image with a 905 nm laser"). I think better terminology could be used, especially when it is clear that these are likely to be isolated single defects.

2. The authors refer to SiC as a "technology-friendly electronic material" but should be more specific what they mean. Presumably the implication is that SiC is more amenable to large scale processing and electronic integration.

3. The pulse diagrams in Fig 3 could benefit from more detail, ie. closer to what is in the supplement as Supp Fig 3.

Version 1:

Reviewer comments:

Reviewer #1

(Remarks to the Author)

The authors have thoroughly addressed all our questions, providing clear explanations that resolve the initial ambiguities. Additionally, new data have been incorporated into the paper to delve deeper into the spin dynamics observed in the PDMR measurements. Overall, the authors' responses are satisfactory, and I am inclined to recommend this paper for publication in Nature Communications.

However, before making a final decision, two points of confusion still require further clarification:

1. In their response to Question 1, the authors suggest that higher doping densities can lead to increased background currents. However, in Ref. 10 of the main text (Nat. Commun. 10, 5569 (2019)), the surface in contact with the electrodes is doped with a very high density of nitrogen ($8 \times 10^{17}/\text{cm}^3$) to adjust the Fermi level of silicon carbide and form the Schottky contact. The formation of a Schottky contact is known to significantly reduce background current, which appears contradictory to their statement. Could the authors clarify this apparent discrepancy?

2. The authors describe in the main text that gold is used as the electrode material to form a Schottky contact with the silicon carbide surface. However, gold has poor adhesion to silicon carbide. Is it not necessary to first coat the surface with an adhesive layer, such as titanium, to ensure stability? Further explanation would be appreciated.

Reviewer #2

(Remarks to the Author)

The authors provided a revised manuscript and responded in detail to some of the review queries, as well as provided amendments to the manuscript.

Still several questions asked in the first review round remained unanswered or answers of are unsatisfactory. This prohibits any publication in the current m/s state.

Although a single spin qubit readout in SiC is of high interest to the community, it is not discussed here in terms of new physics but technical improvements, leaving still a quite large number of unanswered questions. As a consequence the paper, as written now, is not suitable for NatureComm.

The photoelectric detection in SiC was reported in the referred publication 10 and the original methodology was developed in previous papers such as those on diamond and properly cited. But, there is no new physics discussed here in a major way (though there is a space for it).

The main problem is as follow (mentioned in the first report). The photoelectric readout in diamond was based on rate dynamics between defect photoionisation from the ES triplet to E_c and the MS transition for spin 1 system with a low spin-orbit interaction. The SiV defect is a $1/2$ - $3/2$ system with two orbital branches. The MS state, on which the photoelectric readout in publication 10 is based upon, is an unconfirmed hypothesis. Could the authors experimentally directly confirm that there is a MS state for this system and how it looks like (symmetry, spin number) ? Then yes the paper would bring new physics

The authors just take over the older hypothesis and don't do efforts to confirm/deny it, for which the single defect centre would be an advantage. For example the authors could execute bunching experiments as a function of the laser power to try to verify it or similar.

Also the authors hypotheses about the 2 photon (and in the rebuttal letter 4 photon for QE – as they claim that one needs

still two photons to recharge the state), but the data presented in figure 2f show a quasilinear behavior. This also magnifies the above discussed need to understand the photoionization mechanisms, not taking over only hypothesis from the ref 10. Only indirect measurements based on photocurrent values are provided as a supporting argument for the number of photons needed for one charge carrier pair

Also, as mentioned in previous review round, there is a basic misunderstanding about the characteristics of back-to-back Schottky diode characteristics and terminology. Flat band conditions on a junction mean, that a voltage is applied equaling the barrier height with an opposite electric field polarity. The authors hypothesize about the barrier height but do not measure it. If it is about the estimated 1.5 V, then the flat band condition (i.e. forward direction) for one of the junction is 1.5 V, it cannot be 9 V or even 10V. At the flat band voltage the contact becomes injecting.

The depletion region is very different terminology, by increasing the voltage on reversely biased junction one depletes the charge from the region between the contact and locate it towards one electrode. I.e. in a p-doped material the holes will be depleted top a width called depletion region width.

The problems that the authors have back-to-back diode, which can be though easily modelled. On the moment when they put flat band voltage on one electrode, the 2nd junction gets double barrier voltage. The location of the defect center in between the contacts and the field profile is utterly important elucidate the transport mechanism. This has not been done.

Although I would be in great favor if single SiV photocurrent technique would be published it cannot stay solely on technical improvements of previous works and the basic knowledge of device physics on an adequate level needs to be provided. In this sense I did not change mind with respect of the first review. But the potential is there.

Reviewer #3

(Remarks to the Author)

The authors have satisfactorily responded to my comments and questions. They explained that the much higher conductivity of SiC presents unique challenges, creating large background currents that would saturate any high gain amplifier and swamp the tiny photocurrent from single defects. Even if optical lock in was implemented, this would make single defect sensing impossible. This motivates the specific use of Schottky type contacts, and this is one of the things that enables single defect PC measurements. This point is made in the revised manuscript and I think justifies the paper's claims to significance.

I thank the authors for their detailed comments and additional measurements, and note that the addition of Ramsey and Spin echo data really pushes the paper over the line as far as coherent spin control goes. The authors changes to the paper are extensive and thorough, I am happy to recommend publication in Nat Comms.

Version 2:

Reviewer comments:

Reviewer #1

(Remarks to the Author)

We appreciate that the authors have provided satisfactory responses to all of our questions and concerns during the first and second rounds of review. We are pleased to recommend this paper for publication in Nature Communications.

To enhance the impact of the current work, I encourage the authors to provide additional details regarding the methods they employed. Below are some specific suggestions for the authors' consideration:

1. As the authors note, there is small frequency-dependent crosstalk from the RF signal to the detected photocurrent. Since the dynamics of carriers in semiconductors can be influenced by the electromagnetic field of the RF signal, the crosstalk can induce significant noise during PDMR measurements. When using lock-in amplification, the RF amplitude is modulated at the modulation frequency, and the crosstalk can manifest as a large noise signal oscillating at the same frequency. This could severely compromise the measurement. Based on the excellent results presented by the authors, it appears that RF crosstalk is nearly entirely suppressed in this experiment. Could the authors provide further details on the methods they used to suppress RF crosstalk noise?
2. In Figure 1b, the authors provide a side view of the sample and its layered structure. However, the shape and relative positioning of the electrodes and the RF antenna are also critical for efficient photocurrent collection and minimizing RF noise. We recommend that the authors include a schematic diagram showing the vertical view of the fabricated sample surface, illustrating the complete configuration of the electrodes and the RF antenna.
3. As described by the authors, the sample is mounted and wire-bonded onto a custom PCB. Could the authors elaborate on the grounding methods employed in the design of the PCB? Specifically, do the RF circuit, the bias voltage application circuit, and the photocurrent collection circuit share a common ground? The grounding approach is critical for effective noise suppression.

Point-by-point response to the REVIEWER COMMENTS

We acknowledge all reviewers for the review of our manuscript. The answers to the Reviewers are shown below in a point-by-point manner. In this letter, the numbers for References cited are written as ‘reference number after revision (reference number before revision)’ to clearly correspond to the numbers indicated by the reviewers and the numbers before and after revision if the reference number changed in the revision process [for example, Ref. 26 (23)]. The changes in the manuscript and Supplementary Information are highlighted in yellow.

Reviewer #1 (Remarks to the Author):

Reviewer’s comment:

This work demonstrates the photoelectrically detected magnetic resonance (PDMR) measurement of a single silicon vacancy spin in 4H-SiC, wherein the magnetic resonance of a single spin can also be optically detected, known as optically detected magnetic resonance (ODMR). By optimizing the ionization pulse control, the signal-to-noise ratio (SNR) of PDMR is shown to be 1.7-2 times higher than that of ODMR. The underlying mechanism is analyzed based on the transition processes of silicon vacancy electrons under excitation. The demonstrated electrical single spin qubit readout method in integrated semiconductor materials could further extend the application field of spin defects in SiC. However, there are several points that the authors need to clarify before I can recommend publication in Nature Communications:

1. The sample is a p-type 4H-SiC with a dopant density of $2 \times 10^{14} \text{ cm}^{-3}$. Is the dopant density related to the electrical readout efficiency? Would similar results be obtained for an N-doped sample?

Our reply:

We have not tried to measure samples with other dopant densities or n-type samples. A higher doping density may increase the background current, but it may be not very critical when the device is operated above the flat-band voltage because the dopants will be fully depleted in this condition. However, a higher voltage or shorter electrode distance is required to achieve the flat-band condition at a higher doping concentration. Also, the dark current may increase due to the narrower depletion width that may induce the charge tunnelling from the metal to the semiconductor at higher doping densities. We expect that electrical readout is possible with N-doped samples.

Reviewer’s comment:

2. Although no signal was detected when measuring the ODMR spectrum of spot B, is it possible that

the spin of spot B could be read out using the PDMR method? If so, this would represent a significant advantage of PDMR over ODMR.

Our reply:

We thank Reviewer 1 for pointing out the possibility of PDMR of spot B. We have not measured PDMR on spot B. We guess that spot B is probably a V1 centre considering the fluorescence spectral range, and we have not observed V1 PDMR in our ensemble study in Ref. 12. As the spin polarization of V1 is not possible with non-resonant excitation [R. Nagy et al., Phys Rev. Appl. 9, 034022 (2018); N. Morioka et al., Phys. Rev. Appl. 17, 054005 (2022)], we consider that PDMR is not possible at room temperature as well as ODMR.

Reviewer's comment:

3. It would be valuable to measure the Rabi oscillation using both the ODMR and PDMR readout methods under the same microwave power and compare the Rabi oscillation frequencies obtained with these two readout techniques.

Our reply:

We appreciate the reviewer's valuable comment. We measured ODMR and PDMR at the same microwave power, and we obtained the Rabi frequencies of 4.12 ± 0.03 MHz and 4.11 ± 0.07 MHz, respectively. We mention the observation of the same Rabi frequency in the revised manuscript on pages 9 and 10, L243–245, and the measurement details and result are added in the revised Supplementary Information as Supplementary Note 3 and Supplementary Fig. 3.

Reviewer's comment:

4. For the Rabi oscillation shown in Fig. 3(d), it would be appropriate to include a scale bar representing the signal contrast, similar to those in Figs. 3(a), (b), and (c).

Our reply:

We thank Reviewer 1 for the suggestion. We added a scale bar in Fig. 3d to show the contrast of Rabi oscillation.

Reviewer #2 (Remarks to the Author):

Reviewer's comment:

The authors present an interesting work on photo-electrical readout of a single V2 centre spin state in SiC with a potential of using it as a spin qubit. The work appends on previously successfully established photo-electrical readout of V2 ensembles (Ref. 10) in SiC and using the technique developed in Ref. 9 for NV diamond. The successful demonstration of single V2 centre readout is possible to achieve due to the optimisation of the experiments, and the spin initialisation and readout by tailored laser pulses, adopting and improving the technique developed in Refs. 16 and 9. However, although this work is highly relevant for development of the solid state quantum technology and for miniaturisation of quantum devices, it is based on improvements rather than coming with novel breakthrough ideas. Also there are some very basic misunderstandings in semiconductor physics, in otherwise excellently carried out quantum optics experiments, in particular see below.

Our reply:

We are very grateful to Reviewer 2 for reviewing and pointing out the concerns regarding semiconductor physics. While we believe we had already carefully considered the device physics in our original manuscript, we would like to show our gratitude for the reviewers' comments that provided us with an opportunity to revise the manuscript to convey our concept more understandably. To address Reviewer 2's concerns about semiconductor physics, we would like to provide point-by-point responses below. To answer the reviewer's questions and comments, we have split the comments we received into shorter sections and answered each one.

Reviewer's comment:

2-i-1 i. One of the novelty claim is to use to use the Schottky contacts, to avoid to employ what the authors call " Ohmic annealing", but this claim is not significant. For example Schottky contacts have been used in ref. 16 for NV centres, but which type of the contacts is best to be used is just a technical decision to be made for concrete experiment as both (injecting, Schottky) are very standard device configurations.

Our reply:

We appreciate Reviewer 2's comment. Although the Schottky contact is a very standard device configuration in electronics and it has also already been used in Ref. 16 (16) for the PDMR of NV centres, the effectiveness of the Schottky contact for PDMR has not been discussed in previous studies. Instead, Ohmic contact has been employed for the electrical detection study of diamond NV spins in many studies [Refs. 9, 13, 14], including the single NV detection [Ref. 13]. However, we should point

out that using Ohmic contact without the current blocking mechanism would lead to a large background current particularly in the case of SiC, and thus we suppose that the use of blocking device configurations such as a Schottky contact and a p(i)n junction is crucial to realize single-spin detection.

The important difference between diamond and SiC is their electrical conductivity at room temperature. Diamond has extremely high electrical resistance. For example, for the ultrapure diamond with a nitrogen concentration below 0.15 ppb used for PDMR of single NV in Ref. 13, the expected resistivity is $\sim 10^{13} \Omega \text{ cm}$ assuming the nitrogen concentration of 0.15 ppb = $2.6 \times 10^{13} \text{ cm}^{-3}$ and the electron mobility of $4500 \text{ cm}^2 \text{V}^{-1} \text{s}^{-1}$ [J. Isberg et al., *Science* 297, 1670 (2002)]. The CVD-grown diamond can exhibit even higher resistivity as high as $10^{15} \Omega \text{ cm}$ [Ref. 23 (new)]. On the other hand, the expected resistivity of our sample, a p-type SiC with presumably boron acceptor with the density of $2 \times 10^{14} \text{ cm}^{-3}$, is about $(4-7) \times 10^2 \Omega \text{ cm}$ assuming 0.24–0.28 eV acceptor level [M. E. Bathen et al., *Mater. Sci. Semicond. Process.* 177, 108360 (2024)] and the hole mobility of $95 \text{ cm}^2/\text{Vs}$ [T. Tanaka et al., *J. Appl. Phys.* 123, 245704 (2018)]. For samples for ensemble study, with dense defect densities higher than the dopant concentration, charge trapping makes SiC semi-insulating [Refs. 10, 25 (new)]. However, in samples containing isolated single defects, the resistivity of the original epitaxial layer is almost preserved even after electron irradiation. The expected density of the carbon vacancy, the dominant defect in the material, is $1 \times 10^{13} \text{ cm}^{-3}$. As this defect density is only 5% of the dopant density, the defect does not increase the resistivity. Therefore, the resistivity of SiC is ten orders of magnitude lower than that of diamond. If the Ohmic contact were used on SiC, a large background current would inhibit the single spin detection. High-gain transimpedance amplifiers typically have an input-equivalent noise level of $0.2 \text{ fA Hz}^{-1/2}$, which corresponds to the shot noise of the electrical current of 125 fA. Subpicoampere level leakage, therefore, would deteriorate the detection signal-to-noise ratio. This is the motivation to use a Schottky contact that blocks the leakage current from outside the SiC whereas photoionized charge carriers can be efficiently collected.

The aspects discussed above were not well described in the original manuscript. Therefore, to help the readers better understand, we added the information on the difference in resistivity between diamond and SiC and the explanation of why current blocking electrodes such as Schottky contacts are necessary on page 4, L73–77 and on page 5, L95–108. Also, we denoted our sample is presumably boron doped in Methods on page 14, L385.

Reviewer’s comment:

2-i-2 On opposite ohmic contact would lead to the photocurrent gain, so signal can be much higher than for Schottky contacts, so it is matter of choice and it is mainly important how the ionisation/recombination and the charge carrier transport are used in the device.

Our reply:

We thank Reviewer 2 for the comment. We are aware that photoconductors can exhibit photocurrent gain. However, this effect is not intrinsic [Y. Dan et al., ACS Photon. 5, 4111 (2018)] but caused by traps and other defects. To electrically identify a single defect, the material needs to have small defect densities. Also, the gain mechanism based on the extrinsic traps is not selective to the detection of the target defect. Therefore, we do not anticipate exploiting extrinsic photocurrent gain by using Ohmic contacts, which may greatly sacrifice the sensitivity because of a larger leakage current. Therefore, for single defect detection in silicon carbide, we believe that it is not a simple matter of choice which type of contacts need to be designed.

Reviewer's comment:

2-i-3 Also annealing is used in the special case of diamond to engineer ohmic contacts by forming a charge carrier tunnelling barrier through 10 nm thin TiC, produced from Ti during the annealing. However even without annealing, TiC can be formed on diamond but the layer is thinner. So annealing is not a must and avoiding it is no significant idea which would make the paper more impacting. SiC is different material than diamond and with significantly lower gap so ohmic contact forming is easier but even though for some Schottky contacts on SiC the annealing is necessary. But these are technical aspect which can be used in Supplementary. Also the contact formation using Au is poorly documented.

Our reply:

We thank the Reviewer 2 for the comment. We do not intend to emphasize in the manuscript that avoiding annealing is a significant idea in our research. However, we would like to note that forming an Ohmic contact on SiC is not easy. Ohmic contact formation on SiC usually requires annealing at 900–1000°C [Ref. 24 (new)], at which temperature silicon vacancies anneal out. Also, to obtain Ohmic contact, the doping density in SiC needs to be increased. Reviewer 2 pointed out that some Schottky contacts on SiC require annealing, but for Au Schottky, annealing is not necessary as reported in Refs. 27 (Sup. Ref. 1) and 28 (Sup. Ref. 2). Regarding Reviewer 2's comment that the contact formation using Au is poorly documented, the fabrication process of the device, including the formation of the Schottky contact, is written in Methods (L260–265) in the pre-revision manuscript. In the revised manuscript, we added details of the processing. Our Schottky contact was formed by the following steps after depositing PECVD SiO₂ and the bonding pad formation: (1) exposure of the SiC surface by BHF etching of SiO₂ with a photoresist mask, followed by the photoresist removal by organic solvents (2) photolithography of the contact electrode pattern, (3) Au film deposition by using the electron-beam evaporation technique, and (4) lift-off. We did not perform annealing after the contact formation. We added these detailed processes in the revised Methods on page 15, L397–404.

Reviewer’s comment:

2-ii-1 ii. There are significant questions concerning device physics. The authors state that they operate Schottky diode and reach the flat-band conditions at 9.1V, for which they design contact geometry, taking into the account the background acceptor and hole concentrations. They also present in SI IV characteristics to support this statement. However, the situation is much more complex and basics devices physics, to which the authors refer, is not well interpreted. Referring to Fig 1, the authors have no single Schottky diode but two of them back to back connected, i.e. one of two (in dark) is always blocking. If the flat band conditions occur at some voltage V_b (equal to barrier height) on one Schottky diode, the potential on the other diode is equal to two barrier heights. So one can not speak globally about the flat band conditions in this situation where there is no uniform potential across the sample throughout the gap between the contacts (the second contact has a double heigh barrier).

Our reply:

We would like to thank Reviewer 2 to raise this question. We agree that our explanation of the flat-band condition is not clear enough in the manuscript. As Reviewer 2 pointed out, our device is a metal-semiconductor-metal structure, which is a back-to-back series of Schottky contacts. The flat-band condition in this configuration is discussed in a semiconductor textbook cited as Ref. 26 (23) and a theoretical modelling study cited as Sup. Ref. 1 (Sup. Ref. 3). As discussed in these references, the energy band is not uniform throughout the semiconductor layer, but the energy band at the forward-biased contact edge becomes flat at the flat-band voltage. At this voltage, the energy band at the other contact is bent by the reverse bias. In this condition, in our p-type material, the barrier for the holes transported from the semiconductor to the metal becomes zero, which realizes near-unity hole collection. Electrons are transported toward the other contact, which is effectively forward-biased and thus there is no barrier for the electrons transported from the semiconductor to the metal. Therefore, both electrons and holes generated from the defect can be efficiently collected by the electrodes, which is very important for detecting the PDMR signal efficiently. In contrast, the electrons and holes from the metal to the semiconductor are blocked by the Schottky barriers. To make it easier for readers to understand, we added these explanations about the carrier transport in the MSM structure on page 5, L113–119; “In the case of our p-type ... contributes to background noise.”. Also, to clarify the role of the charge blocking from the metal to the semiconductor, we changed a sentence in L121–122 [original: “to achieve objectives such as leakage current reduction”; revised: “to achieve leakage current reduction”].

Reviewer’s comment:

2-ii-2 Secondly, the authors misinterpret the term of flat-band conditions and potential for the full

depletion (which are tow very different terms). Flat band conditions, as stated before, occur if the applied voltage in forward direction is equal to the barrier height. As the barrier height for Au on SiC is about 1.5 V. The flat band potential must be the same (can be slightly influenced by the interface state charging). So flat band conditions can not simply be 9.1 V. the authors misinterpret this term in place of stating depletion.

Our reply:

We appreciate Reviewer 2 for mentioning the full-depletion condition. The full-depletion condition is indeed different from the flat-band condition. The flat-band condition described above is not achievable at the voltage corresponding to the Schottky barrier height (1.5 V in Reviewer 2's comment) but is achieved at the voltage even above the full depletion voltage (referred to as the reach-through voltage V_{RT} in Sup. Ref. 3 (Sup. Ref. 1)). At small voltages, the majority of the voltage drop occurs to expand the depletion layer at the reverse-biased contact. As increasing the voltage, two depletion layers at both contacts touch each other, which is the full-depletion condition. At this voltage, the energy band at the forward-biased contact is slightly bent and still forms a barrier for the holes. By applying the voltage further, this barrier at the forward-biased contact reduces, and finally, the energy band becomes flat at the forward-biased contact, where the flat-band condition is achieved. To clarify that the flat-band condition is different from the full-depletion condition, we add a sentence on page 5, L111–113; “The flat-band condition is ... above the full-depletion condition.”

Reviewer's comment:

2-ii-3 Also for voltages above the flat band the photocurrent saturation cannot be achieved as the contact becomes injecting, but the saturation is due to the full collection and to the fact that back to back (ie always in reverse) Schottky diodes are used. Actually the barrier would becomes injecting i.e.will be ideally Ohmic in flat band conditions. These are quit basics semiconductor physics terms which need to be put right.

Our reply:

We appreciate the comment. At the flat-band condition, the electrical current injected from the metal to the semiconductor is governed by the reverse-biased Schottky contact. As both contacts at two metal-semiconductor interfaces are in the blocking configuration in terms of the charge injection from the metals to the semiconductor, the charge injection to the semiconductors can be reduced by choosing a metal that forms large barriers both for the conduction band and valence band. This is a key point of using the Schottky contact to reduce the dark current. The electron-hole pair generated inside the semiconductor are collected by two different electrodes which are both forward-biased for each polarity of charges, as described in the reply to comment 2-ii-1 of Reviewer 2.

Reviewer's comment:

2-ii-4 also, the IV characteristics in the supplementary is measured only in one direction and should be measured in both directions to diagnose in which model the device actually operates. As mentioned, the fact that the author see the photocurrent saturation means that the device operates in non-injecting, i.e no flat-band condition. However the claim that the collection efficiency is one (which can not again be in the flat band conditions), taken from basics semiconductor book is not convincing. This assumes no recombination.

Our reply:

We thank the reviewer for the comment. As our voltage source is unipolar, we measured only in one direction. However, as pointed out by Reviewer 2, this device is a symmetrical back-to-back Schottky configuration. Therefore, the $I-V$ characteristics are expected to be symmetric. As explained in the reply to Reviewer 2's comment 2-ii-1, above the flat-band voltage, no potential barriers exist for both electrons and holes generated inside the semiconductor, which is the reason for the observation of the saturation of the photocurrent.

Regarding the recombination, as we have discussed in the third paragraph of Supplementary Note 1 before the revision, the recombination is considered negligible in our material because the expected carrier lifetime (about 1 μ s) is much longer than the carrier transit time on the order of nanoseconds. This comment is related to the comment 7 of Reviewer 3. To address the consideration of the recombination in the main text, the discussion on the charge recombination previously in Supplementary Note 1 is moved to the revised main text with additional information on page 6, L133–147.

Reviewer's comment:

2-ii-5 To claim such high collection efficiency, the authors need to precisely characterise the junction, (measure both barriers) show the mentioned CV plots (but they are not presented and should be both in dark and under illumination using different laser power) to provide full device characteristics. The IV characterises in Si do not point to such high collection efficiency (and should be fitted with the diode factor).

Our reply:

We thank Reviewer 2 for the comment. Although the measurement of both barriers is suggested by Reviewer 2, the device is symmetric. Therefore, we believe that unipolar measurement is enough to characterise the device, although our measurement polarity is limited by the unipolar voltage source.

Although the $C-V$ plot is suggested by Reviewer 2, the expected capacitance is several femtofarads due to a low doping density and a small contact area. Also, a parasitic capacitance from the bonding pad is added, whose area is about 200 times larger than the Schottky contact. Therefore, it is difficult to measure the $C-V$ characteristics of the Schottky contact of our device. However, the information extractable from the $C-V$ plot is the barrier height and the doping density. The doping density was pre-characterized on the epitaxially grown wafer by the $C-V$ measurement, as we have written in Methods in the previous version. Also, the value of the barrier height does not affect the flat-band voltage in the symmetric device. For these reasons, we did not perform the $C-V$ measurement on the fabricated device. Regarding the comment on the $I-V$ characteristics, the objective of showing the $I-V$ curve in Supplementary Figure 1b is not to characterize the Au/SiC contact, but to show that the device exhibits sufficiently low leakage current to measure a small PDMR signal. We do not think that the fitting of the $I-V$ characteristics will provide pure information on the Au/SiC contact itself because the current should contain a parasitic leakage current of SiO₂ and/or at the surface of SiC. To make this viewpoint clearer, we added the note for the $I-V$ curve in Supplementary Note 1 that the measured current may contain the parasitic leakage current of SiO₂ and/or at the surface of SiC (Supplementary Information, page 1, L28–31).

We also note that characterising $I-V$ and $C-V$ of the Schottky contact is not straightforward in our sample because of the p-type epitaxial layer grown on an n-type wafer, which inhibits the vertical current flow. To obtain a good Ohmic contact on the low-doping p-type epitaxial layer, additional Al ion implantation and high-temperature activation annealing (1600–1800°C) are necessary. However, the properties of the Au/SiC interface have already been studied well in other literature [Ref. 27 (Sup. Ref. 1) and Ref. 28 (Sup. Ref. 2)]. Because we obtained a very small dark current (\sim fA) including the parasitic leakage, we believe that the Au/SiC contact on our device is operating as the Schottky MSM device.

Reviewer’s comment:

2-ii-6 Also the authors might forgotten in their description about the fact that both electrons and holes i.g. a charger carrier pair is generated from V2. This means that one type of the photo carrier can in reality can cross one of the barriers, while the second polarity of the charge the second barrier. This should be implemented in the models. This could bring to my opinion bring the paper needed innovation and allow to discuss the full photoionisation and transport cycle in the benefit of justification of the publication in Nature Comm.

Our reply:

As we explained in response to the comments 2-ii-1, 2-ii-2, and 2-ii-3 from Reviewer 2, our model

considers the fact that both electrons and holes are generated and are collected separately by two different electrodes without barriers.

Reviewer's comment:

2-Iii-1 Iii. The authors mention that one photoionisation cycle generates current of one elementary charge. This is not precise it is one electron-hole pair which is generated.

Our reply:

We thank Reviewer 2 for pointing out this issue. As pointed out by Reviewer 2, one photoionisation cycle generates one electron-hole pair (two elementary charges). However, it should be noted that the electrical current measurable from this pair is only one elementary charge because of the following reasons. Although both electron and hole are transported inside the semiconductor, the movement of the electron in the external circuit is only one elementary charge. The hole, i.e. one electron deficiency is compensated by the electron. As a result, the electrical current measured as a circuit corresponds to one elementary charge, not two. This mechanism is the same as in normal photodetectors. To clarify this, we made a minor revision in the main text on page 11, L301: (before revision) “one photoionisation cycle generates a current of one elementary charge”; (after revision) “one photoionisation cycle generates a current corresponding to one elementary charge”.

Reviewer's comment:

2-Iii-2 However the discussion of the photoionisation efficiency is unclear. The author state that ionisation cross section is about 50 times smaller than the absorption cross section. Which absorption the authors exactly mean, the ZPL absorption?

Our reply:

We thank Reviewer 2 for pointing out that the meaning of the absorption cross-section is not clear. This is not the ZPL absorption but the total absorption at 905 nm to excite the system from the ground state to the excited state. We have included this information in the pre-revised Supplementary Note 5, L173–174 as “Therefore, the ionization rate is about 50 times slower than the excitation rate at 905 nm”. However, this information was not clearly written in the main text. Therefore, we changed the sentence in the main text on page 13, L343–346, to “As the ionisation cross-section is found to be about 50 times smaller than the absorption cross-section from GS to ES at 905 nm, the PDMR could be further improved by ...”, by adding the phrase “from GS to ES at 905 nm”.

Reviewer's comment:

2-Iii-3 Finally if the second step, i.i. photoionization is 50 times smaller than the first step, it means

that 50 x more photons are needed for this step. How then the current collection can be (with respect of incoming photon numbers) 40% ? In the full rate equation model it means, that due to this 1:50 ratio, only 1 electron of 50 is used for the photoionisation.

Our reply:

We appreciate the reviewer's question. Indeed 50 times more photons are needed to have the same probability for the ionisation from ES and the excitation from GS to ES the same. However, we would like to note that one ionisation event absorbs only one photon, not 50 photons. Therefore, it is not the case that only 1 electron out of 50 is used for photoionisation. The defect can be excited with a unity probability if the laser power is strong enough. Our discussion on the photoionisation efficiency of 40% means that the probability of successful photoionization per laser pulse is 40% with the nanosecond laser pulse, whose power is strong enough to compensate 50 times less efficient ionisation cross-section compared to the excitation from GS to ES.

Reviewer's comment:

2-Iii-4 This can be in general expressed in the term of QE for PL and QE for photocurrent which have to be compared (as a function of the laser power). These numbers and consequently the theoretical S/N ratio do not match. The authors need to express the number of generated terms in the framework of QE. This situation is different to diamond where the QE for PI is about 80%, While QE for photocurrent is about 50%. For the case of V2 in SiC, based on the author rate comparison it seems to be much lower, and authors need to clarify it.

Our reply:

We appreciate the reviewer's insightful comment. Unfortunately, the value of the quantum efficiency (QE) of the photocurrent of diamond (50%) mentioned in Reviewer 2's comment is not very clear in terms of the definition of QE. However, we explain our understanding of the quantum efficiency as follows. For photodetectors, QE is usually defined by the ratio of the number of generated electron-hole pairs to the number of incident photons. This value should be dependent on the excitation power and duration for the ionization of the single defects, but the QE for V2 calculated with this definition is about 6×10^{-11} with a laser pulse at 35 mW for 39 ns. On the other hand, the QE of the photoluminescence is usually defined by the number of emitted photons per absorbed photon. When the QE of the photocurrent from the single defect is similarly defined as the number of electron-hole pairs per absorbed photon, it is estimated to be about 0.14 with the laser pulse of 39 ns at 35 mW (about two chances of GS→ES excitation before ionization or transition to MS on average, 0.4 photons for ionization, 0.4 photons for recharging: $0.4/(2+0.4+0.4)=0.14$). At infinitely high laser power, the QE of photocurrent will be 1/3 for the V2 centre (one photon for excitation from GS to ES, one photon

for ionisation, and one photon for recharging: $1/(1+1+1)=1/3$). With this definition, the QE of the photocurrent of the NV centre at infinitely high laser power will be 0.25 instead of 0.5, because four photons are necessary to generate one electron-hole pair (one for excitation, one for ionization, two for recharging). Because this definition involves the number of photons absorbed for excitation, it is not easy to understand the probability of having the ionisation event. Therefore, we described the ionization probability (efficiency) of 40% in the manuscript.

Reviewer's comment:

2-iv-1 iv. There are still some other technical issues. The methodology for the contact preparation needs to be described as the surface plays in this process an important role and contacts need to be characterised. Also in Supplementary the photo I-V is shows the signal less than the dark current, is the dark current from the signal case subtracted, please clarify.

Our reply:

We are grateful to Reviewer 2's comment. We have clarified the preparation method of the contact in response to comment No. 2-i-3. Regarding the comment on photocurrent–voltage measurement shown in Supplementary Figure 1, we do not subtract the dark current in the data. As the photocurrent is measured with lock-in detection by modulating the laser intensity, the dark current, the DC component, does not appear in the result of the lock-in measurement. To clarify this fact, we added information that the photocurrent is measured “by modulating the laser intensity” in the figure caption of Supplementary Figure 1a.

Reviewer's comment:

2-iv-2 Further on, there are a few important details concerning the detection. The authors use the amplifier with the amplification $10E11$, for this the bandwidth is typically close to 1 second, the authors should clarify/put reference to methodology how they can measure so fast signal after the illumination with laser pulses on 40 nanosec (or 1000 nanosec) range and Rabi period in the range of 100 nanosec.

Our reply:

Thank you for your comment. As we wrote in Methods, our signal bandwidth is 100 Hz. We used a 10^{12} V/A transimpedance amplifier (NF SA-609F2) whose bandwidth is 300 Hz. After the transimpedance, the signal is low-pass filtered at the cut-off frequency of 100 Hz. Also, we wrote in L310–312 of the original main text by citing Ref. 63 (47) that “we repeat short pulse sequences and modulate the laser amplitude with a slow square envelope at 7 Hz for lock-in detection⁴⁷.” As discussed in Ref. 63 (47), fast photocurrent can be encoded to a slow envelope signal by modulating the laser

intensity within the bandwidth of the slow detection electronics. To visualise this pulse-sequence encoding in the slow lock-in reference, we added the illustration of the pulse sequence in the revised Fig. 2e.

Reviewer's comment:

2-iv-3 Also in their pulse sequence the RF is applied after 1000 nanosec after the laser pulse. what is the coherence time of V2 in that case? These are very important parameters to list if one discusses the potential quantum technology applications.

Our reply:

We thank the reviewer for the comment on the pulse sequence. The 1000 ns wait time after the laser pulse is inserted to let the system decay to the ground state to complete the spin-state initialization. In Ref. 18, 600 ns wait time was used. During this wait time, the spin does not hold coherent information, and the spin polarization decays by the longitudinal spin relaxation time T_1 . As the T_1 of single silicon vacancies is reported to be hundreds of microseconds at room temperature in Ref. 18, the decay of the polarization during 1000 ns is negligible. The coherence time is indeed important for quantum technology applications, and the lower limit of the coherence time T_2 of the single silicon vacancy is reported to be 160 μ s in Ref. 18 This question from Reviewer 2 may be the question to the performance of the silicon vacancies in SiC itself, which is out of scope of this study. However, we note that the T_2 time of ensemble silicon vacancies measured by ODMR and PDMR is not different as reported in Ref. 10.

2-iv-4 From the comments above, the paper is by far not ready for any publication (and has basic device physics errors). However I estimate that it can be improved by the full rate modelling using the electron-hole pair generation, drift and the collection by the electrodes and back to back Schottky description .

Our reply:

We sincerely appreciate the comment on how we can improve the manuscript. In response to the comments above, we have explained the device physics. Also, we have explained how we considered the electron-hole pair generation and the mechanism of efficient collection by back-to-back Schottky contacts. We hope the answers above help Reviewer 2 understand better, and we thank Reviewer 2 again for giving us the opportunity to improve our manuscript to clarify these aspects.

Reviewer's comment:

2-iv-5 Also the improvement in SNR compare to ODMR is minor (it can be improve by a SIL by a

factor 50) and the authors should discuss the levels to which it could be enhanced for example in diamond this ratio is about 1000).

Our reply:

We thank the reviewer for the comment. If we try to understand the comment of Reviewer 2 correctly, it is stated that the SNR can be improved by a factor of 50 by using a SIL, which corresponds to the improvement of the photon collection by a factor of 2500. Otherwise, Reviewer 2 might have intended the enhancement of the photon collection factor of 50 by SIL. However, we are afraid that it would be unrealistic to improve the photon collection by even a factor of 50 by a SIL. It is reported that a photon count rate of 0.35 Mcps from NV centres in bulk diamond is observable with a confocal microscope with a high NA objective lens (1.35) [M. Jamali et al., Rev. Sci. Instrum. 85, 123703 (2014)], and 17.5 Mcps (0.35×50) exceeds the theoretical saturated photoemission rate of a single NV centre 14.3 Mcps calculated using the internal transitions rates reported by L. Robledo et al, New J. Phys. 13, 025013 (2011). High enhancement factor can be obtained if the photon count rate in bulk is underestimated by, for example, using an objective lens with a small numerical aperture or with a not well constructed confocal microscope. The important aspect is how many photons can be collected by using the photon collection enhancement structure, and for NV centres in diamond, it is about 1 Mcps for a solid immersion lens [L. Robledo et al., Nature 477, 574 (2011), M. Jamali et al., Rev. Sci. Instrum. 85, 123703 (2014)], and 2.7 Mcps for a bullseye grating [L. Li et al., Nano Lett. 15, 1493 (2015)]. When one compares 2.7 Mcps to 0.35 Mcps, the expected enhancement of SNR is $2.8(=\sqrt{2.7/0.35})$. For silicon vacancies in SiC, 120 kcps is reported from a single silicon vacancy with a cavity antenna [J. Körber et al., Nano Lett. 24, 9289 (2024)]. Assuming a similar ODMR contrast to us, the enhancement in SNR will be approximately 3.5. It is not easy to address the value of SNR enhancement of 1000 by Reviewer 2 because the reference is not provided. However, the simulation of PDMR of the NV centre in diamond in Ref. 16 reports the enhancement of the spin detection sensitivity of PDMR to be about 3 (instead of 1000) compared to the ODMR assuming 0.1 kcps photon count rate. Our simulation in Supplementary Note 7 (6 in the pre-revision version) suggests that the possible enhancement of SNR is about 5, corresponding to the 25 times photon collection enhancement. If Reviewer 2 intended the photoelectric gain by the number of 1000 stated in Ref. 9, such extrinsic effect by the extra traps may not apply to the samples with low defect densities where single defects are photoelectrically resolvable, as mentioned above.

Reviewer's comment:

2-iv-5 Also it should be discussed how was one can detect the signal to achieve SQL as for photon counting which limits, jointly with a extremely low current detected, real application for single SiC V2 qubits, so limits and improvements need to be clearly discussed.

Our reply:

We thank Reviewer 2 for the comment. It is true that achieving the standard quantum limit (SQL) by electrical detection is not very easy compared to photon counting as Reviewer 2 pointed out. This is because of the nonnegligible input equivalent noise of the electrical amplifier. Providing the method to achieve the SQL in the electrical detection is out of the scope of this study, but an idea to realize an electrical single-shot readout has been mentioned in the Ph.D. thesis by E. O. Glen [University of Chicago (2022)] by using avalanche multiplication of the photoionized charges from the defects embedded in a p-i-n diode structure. However, note that the high signal intensity in PDMR can result in better SNR compared to the photon counting which is limited by the photon collection efficiency even though the noise from the electrical amplifier exists. Reviewer 2 mentions ‘extremely low current detected’, but it is impossible for us and for readers to judge whether the current from a single V2 centre is ‘extremely low’ because the previous PDMR data of single NV centres only show the contrast data and no current intensity data is reported for spin readout experiments. Even with the low current, our study suggests a good SNR in PDMR. The method for the improvement and the expected limit of SNR has been written in L209–212 of the original main text as “further possible improvement in SNR by a factor of three by enhancing ionisation. As the ionisation cross-section at 905 nm is found to be about 50 times smaller than the absorption cross-section, the PDMR could be further improved by searching for the optimum wavelength for efficient ionisation.”. In addition, although this study detects V2, the method we have shown is also applicable to other defects in SiC and other systems, so we hope that the paper will be appropriately evaluated from that perspective as well.

Reviewer’s comment:

2-iv-6 Die to deficiencies in device physics, for the moment, the paper can not be published elsewhere. Also the authors use a model from Ref 10, but this is a tentative model, so rate modelling can be off scale.

Our reply:

We hope that Reviewer 2 has understood the significance of this paper based on the explanations that we have provided so far. Finally, we would like to comment on the appropriateness of the rate model that Reviewer 2 mentioned. The used rate model may not be perfect. However, based on this rate model, our idea to illuminate the ionization laser at the beginning of the laser pulse indeed improved the electrical readout efficiency. In addition, our simulation in Supplementary Note 7 (6 in the pre-revision version) did not show a significant contradiction with the experimental results. Therefore, our study shows that there are no significant inconsistencies with the experiment in the basic framework of the rate model cited from Ref. 10. Therefore, we hope Reviewer 2 could understand that our

experimental results even strengthen the reliability of this ionisation model.

Reviewer #3 (Remarks to the Author):

Reviewer's comment:

Nishikawa et al report photoelectrical detection and coherent control of single V- centres in 4H-SiC. This is a first, as to date photoelectric detection has been applied only to ensembles of defects, though was done for single NVs in diamond some time ago. The paper also examines improvements in signal to noise attainable with photoelectrical detection, and shows evidence of better SNR using PDMR than ODMR by using short, high power pulsed light. The paper reports interesting results that I think will be of interest to the community, but I can't find a compelling case for publication in Nat Comms just yet, and so cannot recommend publication at this time. My concerns are listed below, and if satisfactorily addressed the paper could be acceptable.

1. PDMR in SiC was first demonstrated in 2019, and single-NV photoelectric readout in the same year. While the authors report the first photoelectric detection of single centres in SiC, it is not clear what particular challenge or roadblock was overcome and why PE detection of single centres is now possible: the experiment seems very similar to the previous work (including by the same group, ref 12). Is this related to near-surface charge stabilisation of single defects, as mentioned in the supplementary information? If so this point should be made clearer, or any other reason why single defect PDMR is so much harder in SiC.

Our reply:

We thank Reviewer 3 for raising the question about the particular challenge to realise the electrical detection of single centres in SiC. This is related to the answer to Reviewer 2, but the most critical reason for the difficulty in achieving single defect sensitivity in the leakage current in SiC due to its low resistance at room temperature. This is because of shallow dopant activation energies in SiC. A larger background current increases the noise and saturates the amplifier. At the doping level of $2 \times 10^{14} \text{ cm}^{-3}$ (about 4 ppb) which we used for this study, the electrical resistivity is 20 Ωcm for n-type and 200–700 Ωcm for p-type (depending on the acceptor impurity). On the other hand, diamond is almost insulating. At the same nitrogen concentration of $2 \times 10^{14} \text{ cm}^{-3}$ (about 1 ppb), the expected resistivity of diamond is $4.9 \times 10^{12} \Omega\text{cm}$. In previous studies for ensemble spin detection in SiC, the sample was highly irradiated, and thus the sample's resistance increased. From the data given in Ref. 10, we estimate the resistivity of the sample in Ref. 10 before and after irradiation to be 10 Ωcm and mid- $10^6 \Omega\text{cm}$ range, respectively. Thus, the irradiation increased the resistivity by five orders of magnitude. Nevertheless, a nanoampere level dark current was reported in Ref. 10 in SiC, and the resistivity is still at least five to six orders of magnitude lower than diamond. For these reasons, establishing a detection device that can achieve low dark current and high charge collection efficiency

was crucial. As this point was not clearly stated in the original manuscript, we added sentences that explain these issues in SiC in the revised manuscript on page 4, L73–77 and on page 5, L95–108. In addition, the detailed information on the Schottky barrier and the device structure to achieve low dark current, which was written in Supplementary Note 1 previously, has been moved to the main text on pages 6 and 7, L122–131.

Reviewer’s comment:

2. Much emphasis is put on the improvements possible with PE detection and advantages over conventional optical detection. The supposedly enhanced SNR (something like 60x for PE compared to optical according to fig 4) is not at all evident in Fig 2a,d, I would say the confocal optical measurement is far higher SNR. Given the P^2 two photon ionisation dependence, the PE detection features should be smaller than those in standard PL. While the signal to noise is well defined for the spin measurements, how is noise defined in the imaging measurements - is background photocurrent the noise? Why is PDMR in Fig 2 so much better than in Fig 3, when the higher SNR scheme is presumably used in the latter?

Our reply:

We thank Reviewer 3 for the question on the definition of the signal and noise in Fig. 4a. The definition of signal and the noise is defined as follows. We performed the measurements of photon counts and photocurrents at defect A and at a background position to measure the real photocurrent signal (as a mean value) and to estimate the noise (standard deviation). Here, for a fair comparison, the photon count is measured with a 62.5 ms time bin, and the photocurrent value is read out at the same interval from the lock-in amplifier. This interval was selected as a close value used for optical imaging in Fig 2a (50 ms per point), and the data sampling interval in Fig. 2d for electrical imaging (70 ms). We added the explanation of the imaging SNR measurement in the main text on page 10, L264–269. Also, as the laser power and the integration time for optical imaging in Fig. 2a and the sampling interval for Fig. 2d were not written, we added the information to the caption of Fig. 2a (L677) and 2d (L682) in the revised manuscript. In Fig. 4a of the original manuscript, the noise at the defect was included in the noise calculation. However, it is better to use the noise only in the background in the case of imaging. Therefore, we recalculated the imaging SNR values and updated Fig. 4a, with an added explanation of the evaluation method of SNR explained above. In the updated Fig. 4a, the maximum enhancement of photocurrent imaging SNR to photodetection imaging SNR within the measured laser power range is >4.3 instead of >5.5 in the previous version. The photoluminescence image in Fig. 2a was measured at 30 μ W with a 730 nm laser ($SNR \approx 10$), and the photocurrent map image in Fig. 2d was measured at 4 mW with a 905 nm laser ($SNR \approx 20$). We compare the SNR of defect A, whose signal is weak in Fig. 2a but is strong in Fig. 2d. At higher laser power, a more intense photocurrent signal improves imaging

SNR, whereas the photon count rate saturates. Regarding the spot size, one needs to consider the saturation of the transition from GS to ES and the excitation laser wavelength difference. At the laser power of 4 mW, the power dependence of the photocurrent is in the linear region, as shown in the revised Fig. 2f (originally Supplementary Fig. 5, moved to the main text in response to comment 6 from Reviewer 3). The observed linear increase means that the excitation from GS to ES is saturated at this power, and the overall photoionization dynamics are limited by the ionisation rate from ES to the conduction band. Therefore, the photocurrent imaging resolution is determined not by the square of the input Gaussian beam (diameter/ $\sqrt{2}$), but by the input Gaussian beam size (diameter \times 1). This discussion on the imaging spot size is added in the revised manuscript on pages 7 and 8, L178–198, and in the revised Fig. 2g. As the laser spot size is larger for longer wavelengths, the imaging spot size is larger in photocurrent imaging. Regarding the SNR of Fig. 2e, the integration time is six times longer in Fig. 2e compared to Fig. 3. With a shorter integration time, the data in Fig. 2 has more noise.

Reviewer's comment:

3. I think the enhanced SNR the authors mention so much would need to be put to the test in a more useful setting than just PDMR. I don't think Rabi alone quite cuts it for demonstrating coherent control, and if SNR can be enhanced so much why are measurements like Ramsey and Hahn echo not done? Also, the authors say on lines 148-149: "Despite that the signal decreases with the two-step laser pulse by 40%, the baseline current decreases ten-fold": I don't see that in the data, has it been subtracted away?

Our reply:

We are grateful to Reviewer 3 for suggesting additional measurements. In previous studies of PDMR, the coherent readout has been demonstrated only with the measurement of Rabi oscillation. The detection of Rabi oscillation demonstrates the ability to detect quantum superposition state, and the previous study of single NV detection published in Science [Ref. 13] also advocates the coherent readout by Rabi measurements. Meanwhile, the reviewer's suggestion is also valid, and thus we added Ramsey and Hahn echo measurements in the revised Fig. 3f and 3g. The explanation for the measurements is added to the main text on page 10, L247–259 and to Methods on page 16, L443 (magnetic field strength for the Ramsey and Hahn echo measurements) and on page 17, L464–471 (detailed method). Regarding the question on the baseline current, it is subtracted away from the data. However, from the value of contrast and intensity, the readers can calculate the subtracted background current. However, to clearly show the background current in the figure, we updated Figs. 4a–c to show the data including the background current without any correction, and we updated the figure caption accordingly (page 28, L700–701).

During the preparation for the measurements for Ramsey and Hahn echo, we found that the spin properties of defect C are slightly different from the V2 centre. However, the spin parameters and the fluorescence spectral range of defect C are very close to those of the V2 centre, suggesting that defect C is a kind of modified silicon vacancies [Ref. 41 (new)]. To compare the spin properties of defects A and C, the PDMR spectra at zero applied magnetic field and at 57.4 G of defects A and C are added in Supplementary Note 5 (page 6, L153–163, and Supplementary Fig. 5. Also, the change related to defect C is implemented in the main text on page 11, L275–277, and in the figure caption of Fig. 4b on page 30, L725.

Reviewer’s comment:

4. The discussion and conclusion sections appear to state that electrical detection is somehow simpler than optical measurements: "PDMR does not require extensive fabrication", "efficient detection with PDMR requires only electrode fabrication" - I really don't buy this, electrical detection is far more of an involved process than optical detection and this paper doesn't convince me otherwise. I think the authors should instead highlight the outlook their technique offers for integrated devices, and why single defects represent a significant advance.

Our reply:

We would like to thank Reviewer 3 for the suggestion to improve the manuscript. Regarding the simplicity of the PDMR, we intended that the PDMR device fabrication could be simpler compared with fabricating photon collection enhancing structures such as solid immersion lenses or photonic cavities. However, both techniques involve fabrication processes. Therefore, we removed the statement pointed out by Reviewer 3 (L229–232 on page 9 in the original manuscript file: “, while PDMR does not require ...miniaturisation.”), and highlighted the outlook that single spin detection in PDMR in SiC is important for enabling integrated and miniaturised quantum information high spatial resolution quantum sensing devices, on page 14, L367–370.

Reviewer’s comment:

5. The authors mention being limited by unknown electrical noise to almost twice their amplifier noise level, but do not suggest the possible sources of this noise or outline mitigation strategies: is this rf rectification related, and that is why the B field is swept rather than the rf freq?

Our reply:

We appreciate the reviewer’s valuable comment. We estimate the possible major noise source is the laser. By blocking the laser illumination to the sample, we observed a decrease in the noise by $0.23 \text{ fA}/\sqrt{\text{Hz}}$. The technical noise from the electronics controlling the experiment is estimated to be 0.20

fA/ $\sqrt{\text{Hz}}$. We expect that the noise of the laser can be mitigated by stabilising the laser intensity. Under the magnetic field sweep condition, we estimate the noise from RF to be 0.1 fA/ $\sqrt{\text{Hz}}$. In the revised manuscript, we mentioned these noise components and the laser for the possible major noise source on page 11, L291–295. The random noise component from RF is small, but RF induces an offset to the signal, whose level is RF frequency dependent. In RF sweep, this offset makes the estimation of the random noise complicated. Therefore, we used the magnetic field sweep. However, PDMR measurement is possible with RF frequency sweep, as shown in Fig. 2e. However, we did not mention that the background was subtracted from Fig. 2e, and thus we added this information to the figure caption of Fig. 2e.

Reviewer’s comment:

6. The results of the simulations detailed in the Supplement are quite tersely reported in the main text and difficult to follow (lines 208-212): did the simulation predict a 40% ionisation efficiency or did it use this result from the experiment? Is the suggested procedure to "enhance the ionisation" equivalent to finding a better wavelength, as is mentioned soon after? The authors might also consider citing Todenhagen and Brandt arXiv:2307.11830 here. I think the paper could also benefit from moving some of the details in the supplement into the main text.

Our reply:

We thank Reviewer 3 for the helpful comment. We first mention the estimation of the ionization efficiency of approximately 40% in L196–201 on page 8 in the pre-revision manuscript. This estimation is based on the repetition rate of the PDMR pulse sequence and the PDMR contrast. The sequence repetition rate of the PDMR in Figs. 3a–c and 4b–c is 395.3 kHz. If each sequence unit generates one electron-hole pair, (elementary charge) $\times 395.3 \text{ kHz} = 63.3 \text{ fA}$ will be generated. With the laser modulation, this value is measured as 28.5 fA with the lock-in amplifier. We observe the experimental PDMR contrast of 2.0%, but considering the background current and the photocurrent generated during spin initialization, the intrinsic contrast is estimated to be 5%. Therefore, $0.05 \times 28.5 \text{ fA} = 1.4 \text{ fA}$ is the expected single-spin signal in PDMR when the ionisation takes place at 100% efficiency. Comparing this ideal signal intensity to the experimentally observed intensity (0.51 fA), we detect $0.51/1.4 = 0.36 \approx 40\%$ of the ideal signal. This value of 40% can be attributed to two factors: ionisation efficiency and charge collection efficiency. If the charge collection is the problem, the spin signal intensity should saturate at high laser powers. However, we observe an almost linear spin-signal increase in the experiment as the increase in the laser power, as shown in Supplementary Fig. 6d. This increasing trend in the spin signal indicates that the ionisation efficiency still has room to increase, suggesting insufficient ionisation efficiency rather than the charge collection efficiency. To make these discussions clearer, we added details of the calculation and the interpretation in the revised main text

on pages 11 and 12, L302–311.

In L208–212 on page 8 of the original manuscript, we made a second mention of 40% photoionization efficiency. This value is estimated from the simulation. This simulation is performed using the ionisation rate estimated from the observed laser power dependence of the photocurrent from the V2 defect under the quasi-continuous-wave laser excitation (discussed in the original Supplementary Note 5) with assumption of a unity charge collection efficiency, and the simulation finds the generated charge per laser pulse to be about 40% at the experimental laser power (Supplementary Fig. 6e). Therefore, the second estimation of 40% ionization efficiency is independent of the first calculation based on the sequence repetition rate. The fact that there is no discrepancy in the estimation of ionisation efficiency using the two methods also supports the assumption that the charge collection efficiency is close to unity. The explanation of the independence of two estimation methods and our interpretation of the validity of near-unity charge collection efficiency is added in the revised main text on page 13, L339–342 (“Although this calculation is ... efficiency close to unity is reasonable.”).

According to the suggestion of the reviewer, we added explanations to the main text on how we evaluated the excitation and ionisation rates from the experimental power dependence of the photocurrent, which was previously written only in Supplementary Information. For this purpose, we moved the figure of laser power dependence of the photocurrent from the previous Supplementary Fig. 5 to the revised main Fig. 2f (this figure is also related to comment 2 from Reviewer 3), and we added the explanation for the Fig. 2f in the main text on page 7, L170–178. Then, we added the procedure and the results of estimating the excitation and ionisation rates from the laser power dependence of the photocurrent on pages 12 and 13, L316–335. We also added a brief explanation of the procedure of the PDMR simulation (Supplementary Note 7 (6 in the pre-revision version)) in the main text on page 13, L335–339 (“Using the excitation and ionisation rates ... by solving the rate model.”). In the updated Supplementary Note 6 (5 in the pre-revision version), we added more details in the derivation of the equations to calculate the excitation and ionisation rates on pages 8 and 9, L206–207, L209–210, L215–217 (Supp. Eqs. (8)–(10)), and L218 in Supplementary Information.

In the process of evaluating the imaging spot size in response to comment 2 from Reviewer 3, we refined the data analysis to extract the laser power dependent photocurrent intensity. Therefore, the values on the revised Fig. 2f are slightly updated from the original Supp. Fig. 5. Also, the resulting excitation and ionisation rates in Supplementary Table 1 and the value in L258 in Supplementary Note 7 are updated accordingly. The rates written in the revised main text (L334–335) are updated values. Due to the small change in these rates, we performed the simulation again and updated Supplementary Fig. 6. However, the change is very small. The values of P_0 and α were wrongly written in the original

Supplementary Information. This was a writing error in the text preparation and did not affect the results of the rate calculations.

Regarding the words “by enhancing ionisation”, we meant to search for the optimum wavelength for ionization, as Reviewer 3 pointed out. To make the sentence clearer, we changed the words in L343 from “by enhancing ionisation “ to “if higher ionization rate is achieved”. Also, we appreciate the reviewer for suggesting a reference. We cited Todenhagen and Brandt arXiv:2307.11830 as Ref. 55 (new) to support the idea of the search for the optimum ionisation wavelength.

Reviewer’s comment:

7. The authors mention a preference for Schottky-type contacts in contrast to the oft reported need for ohmic contacts. The reference to the flat-band condition is particularly interesting, and this point alone is of great interest to anyone working in PDMR particularly in materials such as diamond. I would like to see a few extra details included in the paper (or perhaps a more thorough dataset in the Supplement) that explain how 100% charge collection efficiency is obtained or even possible: in the supplement it is suggested that this arises from the carrier lifetime only, however it is now well demonstrated in diamond that charges emitted from single defects can be captured by other defects, eg. Lozovoi et al Nature Electronics 4 (10), 717-724 2021. Other work has explicitly considered the role of charge capture of luminescent and nonluminescent defects in photocurrent measurements, eg. arXiv:2402.07091. In SiC at room temp these effects are convolved with a significant acceptor and free carrier concentration in contrast to diamond, so I am curious as to just how many charges generated by a photoionisation cycle end up in the detection electrodes.

Our reply:

We would like to thank Reviewer 3 for the questions and suggestions. We recognise that intensive study on the capture of charge emitted from single defects. Because of the research relevance, we added the suggested papers to the References as Refs. 29 (new) and 30 (new). In our study, we only consider the carrier lifetime limited by the carbon vacancy ($\sim 1 \mu\text{s}$) for several reasons. First, the carbon vacancy (expected density = 0.2 ppb) is considered the dominant deep level that determines carrier lifetime in SiC [Ref. 31 (Sup. Ref. 4)]. Second, our device is biased above the flat-band condition, and thus the device is fully depleted. Therefore, shallow dopants are fully ionised and free carrier concentration in the device from the dopants is negligible. Charge capture by the ionized shallow impurities is also negligible because the emission rate is fast. Third, the density of defects other than carbon vacancy and shallow impurities is expected to be small. For these reasons, although the charge capture may exist mainly due to the carbon vacancy, it is a much slower process compared to the

charge travelling time in the device, which will not affect the amount of detectable photocurrent significantly. The discussion on the carbon vacancy and the carrier lifetime was previously written in Supplementary Note 1 (the third paragraph of the original version). By adding the discussions on the charge depletion and the negligible trapping and free carrier concentration from the shallow defects, the whole discussion on this topic is moved to the main text on page 6, L133–147.

Because of the negligible charge capture and free carrier concentration in our device, the question of how many charges are generated by a photoionisation cycle and end up in the detection electrodes can be answered in a similar discussion to the answer to comment 6 by Reviewer 3. By considering the sequence repetition rate and the signal contrast, we observe about 40% of the ideal case where ionisation and collection efficiencies are unity. Therefore, the charge collection efficiency is at least 40%, but as the signal increases as the laser power, the signal is limited by the ionisation. The independent simulation assuming the unity collection efficiency finds about 40% ionisation efficiency. Therefore, we expect that the photocarriers are collected at high efficiency close to unity.

Reviewer's comment:

8. The authors mention that under 730nm illumination, no isolated single site photocurrent sources are identified "probably due to a large background from the interband two-photon electron-hole pair generation". While the powers used are high (30mW+), are they really high enough to do two-photon band-to-band excitation, even when in ns pulses and confocal volumes? Are the authors able to provide a reference or formula for this?

Our reply:

We would like to thank the reviewer for the question. We used a 4 mW laser for photocurrent imaging at 730 nm. The two-photon absorption coefficient was reported to be $\beta_{2PA} = 2.2 \times 10^{-13}$ m/W at 730 nm in semi-insulating 4H-SiC [Supp. Ref. 2 (new)]. A theoretical calculation also reports $\beta_{2PA} = 2.0 \times 10^{-13}$ m/W [Supp. Ref. 3 (new)]. In Supplementary Note 2, we added how we estimate two-photon absorption current using these values, and we find that the expected two-photon absorption current is about 0.6–0.7 pA/mW². Although we observe one order of magnitude smaller total photocurrent and two orders of magnitude smaller quadratically dependent photocurrent in the experiment, this analysis suggests that the two-photon absorption may need to be considered at 730 nm.

Reviewer's comment:

9. I'm curious as to the identity of the mystery defect-like feature near the V2 centre. The authors mention possible candidates such as an N-V defect or divacancy, but should also mention the

emission characteristics of these centres as well, i.e. ZPL, PSB wavelengths.

Our reply:

We thank Reviewer 3 for the suggestion. We added the information on ZPL wavelengths of divacancies and nitrogen-vacancy centres in 4H-SiC on page 13, L353–354.

Reviewer's comment:

10. The lock-in frequency is very low for PDMMR - 7Hz in Fig 2 and 13Hz in the methods. The amplifier has bandwidth up to 300Hz at 10^{12} V/A, could the authors explain their choice of such a low lock in frequency?

Our reply:

We thank Reviewer 3 for the question. The amplifier's bandwidth is 300 Hz, but we find smaller noise in the detected signal at lower lock-in frequencies.

Reviewer's comment:

Other minor points:

1. The authors refer to fluorescent features in the PL and PE detection images as "spots" a few times out of context (eg in the abstract, "We also observe photocurrent spots without photoluminescence in the spectrum range of silicon vacancies", on line 112 "We successfully obtain a clean spot-like photoelectrical image with a 905 nm laser"). I think better terminology could be used, especially when it is clear that these are likely to be isolated single defects.

Our reply:

We thank Reviewer 3 for the comment. We changed the terminology to 'defect' or 'single-defect-like features' when the spots are likely to be single defects. By this change, the text before and after has also undergone minor changes to ensure that the meaning makes sense.

Changes:

(Abstract) L29–30

(main text) L153, L157–158, L161, L164, L166, L167, L275, L278, L348, L349–350, L352, L370

(Figure 2 caption) L678, L679

(Figure 3 caption) L697, L704

(Figure 4 caption) L724, L731, L733

(Supplementary Information) L44, L107, L110, L112, L113, L131, L135–136 (title of Supp. Note 5), L139–142, 145–147, L166

Reviewer’s comment:

2. The authors refer to SiC as a "technology-friendly electronic material" but should be more specific what they mean. Presumably the implication is that SiC is more amenable to large scale processing and electronic integration.

Our reply:

We thank Reviewer 3 for the suggestion to specify the meaning of the phrase “technology-friendly electronic material”. We intended exactly what the reviewer presumed. Therefore, we changed the words accordingly in L24–25 (Abstract) from “a technology-friendly electronic material” to “a material amenable to large-scale processing and electronic integration”.

Reviewer’s comment:

3. The pulse diagrams in Fig 3 could benefit from more detail, ie. closer to what is in the supplement as Supp Fig 3.

Our reply:

We thank Reviewer 3 for this suggestion. We added detailed pulse sequences of ODMR and PDMR in Fig. 2e which was originally in Supplementary Fig. 3 (removed in the revised Supplementary Information). Accompanied by this change, we wrote the information on the rise time of AOM used for optical characterisation also in Methods on page 15, L413–414. Also, the explanation of using the same wait time after the laser pulse and the total sequence length is moved to the caption of Fig. 4b, on page 30, L726.

Revision outside the comments from the Reviewers:

- A) On Page 8, L208: The word “excited state (ES)” is changed to “ES”.
- B) In Acknowledgements, the full name of the person to be acknowledged is described and one additional person to be acknowledged is added.
- C) In Acknowledgements, one grant number is changed (JP23K22796), and one funding source (the Future Development Funding Program of Kyoto University Research Coordination Alliance) is added.
- D) On page 15, L419, in Methods, the spelling of “constructes” is corrected to “constructs”
- E) On page 30, L728, in the caption of Figure 4b, the spelling of “photon-shot-noise-limtied” is corrected to “photon-shot-noise-limited”.
- F) The notation of “flatband” is changed to flat-band throughout the manuscript.
- G) The addition of Fig. 2e’s pulse sequence, 2f, and 2g in response to reviewers’ comments required the rearrangement of texts and the change of sizes of other panels in Fig. 2a–d. However, the data for these panels did not change from the previous version.
- H) We added the explanations of the error bars at the end of the captions of Fig. 2 and Fig. 3.
- I) In Supplementary Information, L14, the address of affiliation number 6 was corrected: from “Tsukuba, Ibaraki 305-0801” to “Tsukuba 305-0801”.

Point-by-point response to the REVIEWER COMMENTS

We sincerely thank all the reviewers for their careful evaluation of our revised manuscript and their constructive feedback. We deeply appreciate their valuable suggestions for further improving the quality of our work. In this response letter, we address each of the reviewers' comments point by point. In this letter, the number for References cited is written as 'reference number after revision (reference number before revision)' when the reference numbers are updated in the revision process due to the addition of one reference Ref. 27 and the reordering of the references. Revisions made to the manuscript are highlighted in yellow for clarity. No change has been made in Supplementary Information.

Reviewer #1 (Remarks to the Author):

Reviewer's comment:

The authors have thoroughly addressed all our questions, providing clear explanations that resolve the initial ambiguities. Additionally, new data have been incorporated into the paper to delve deeper into the spin dynamics observed in the PDMR measurements. Overall, the authors' responses are satisfactory, and I am inclined to recommend this paper for publication in Nature Communications.

However, before making a final decision, two points of confusion still require further clarification:

1. In their response to Question 1, the authors suggest that higher doping densities can lead to increased background currents. However, in Ref. 10 of the main text (Nat. Commun. 10, 5569 (2019)), the surface in contact with the electrodes is doped with a very high density of nitrogen ($8 \times 10^{17}/\text{cm}^3$) to adjust the Fermi level of silicon carbide and form the Schottky contact. The formation of a Schottky contact is known to significantly reduce background current, which appears contradictory to their statement. Could the authors clarify this apparent discrepancy?

Our reply:

We sincerely thank reviewer 1 for their thoughtful and constructive feedback on our manuscript. We are grateful for their acknowledgement of our efforts to address the initial ambiguities and reviewer 1's recognition of the additional data provided to deepen our analysis. Reviewer 1's encouraging remarks and detailed review have been invaluable in improving the clarity and quality of our work. Regarding the two points of confusion mentioned, we appreciate the

opportunity to provide further clarification. Below, we address each point in detail to ensure that all aspects of the manuscript meet the expectations.

The Schottky contact on highly doped semiconductors (doping density N) indeed leads to higher leakage current under an applied reverse bias voltage, and our explanation for higher leakage current at a higher doping level does not contradict the previous study, as we explain below.

In metal-semiconductor-metal junction devices, the leakage current under reverse bias is always a problem because one of the two junctions is reverse-biased in either voltage polarity. At high doping densities, the depletion layer width w_d becomes thinner as $w_d \propto N^{-1/2}$, and the surface electric field E becomes larger as $E \propto N^{1/2}$ [Ref. 26, Chapter 3]. Therefore, at metal/highly doped semiconductor junctions, charges can tunnel through the thin potential barrier on the semiconductor surface. At a doping level of $8 \times 10^{17} \text{ cm}^{-3}$, the reverse leakage current mainly by the thermionic field emission mechanism, where thermally excited charges tunnel across the semiconductor barrier, is considered nonnegligible. [Ref. 26, Chapter 3].

Ref. 10 [Nat. Commun. 10, 5569 (2019)] inserts a 400 nm thick n^+ layer ($[N]=8 \times 10^{17} \text{ cm}^{-3}$) between the metal contact and the lightly n-type doped layer. Here, we consider the device for single-defect detection. Therefore, we focus on the device before the electron irradiation reported in Ref. 10, as the irradiation damage to the crystal and the junction in the defect-creation process is considered small for single-defect application. In the device before electron irradiation, Supplementary Figure 4 of Ref. 10 reports a sudden rise of the leakage current to the μA level at $|V| = 20\text{--}40 \text{ V}$ (Supplementary Figure 4 of Ref. 10). As the contact area can be estimated to be $< 10^4 \mu\text{m}^2$ from Fig. 2a of Ref. 10, the leakage current density is estimated to be on the order of $100 \text{ pA}/\mu\text{m}^2$, which is significantly larger than the dark current in our device fabricated on the low doping SiC ($< 0.1 \text{ fA}/\mu\text{m}^2$). Even after the electron irradiation, the leakage current is 5 nA ($> 500 \text{ fA}/\mu\text{m}^2$) in Ref. 10 at the voltage for their PDMR measurements (20 V). In low-doping semiconductors, the wide depletion layer width prevents the charge from tunnelling, which limits the current flow mechanism to the thermionic emission without tunnelling. Therefore, the leakage current can be reduced greatly by choosing a metal that can form high Schottky barriers.

Reviewer's comment:

2. The authors describe in the main text that gold is used as the electrode material to form a Schottky contact with the silicon carbide surface. However, gold has poor adhesion to silicon carbide. Is it not necessary to first coat the surface with an adhesive layer, such as titanium, to ensure stability? Further explanation would be appreciated.

Our reply:

We are grateful to reviewer 1 for the question. Our Schottky contact is the direct contact of gold to the silicon carbide. As the reviewer pointed out, the gold may not adhere to silicon carbide as strongly as metals such as titanium and chromium. However, we did not have difficulty fabricating and measuring the direct gold/SiC contacts. On the other hand, as we wrote in Methods (page 15, L414–415), we inserted chromium as an adhesive layer between gold and SiO₂ for the bonding pads and the RF antenna because wire bonding is considered to require strong contact adhesion. To clarify chromium's role, we added a sentence, “Here, Cr is inserted as an adhesive layer for stable wire bonding.”, in Methods on page 15, L415–416.

Reviewer #2 (Remarks to the Author):

Reviewer's comment:

The authors provided a revised manuscript and responded in detail to some of the review queries, as well as provided amendments to the manuscript.

Still several questions asked in the first review round remained unanswered or answers of are unsatisfactory. This prohibits any publication in the current m/s state.

Although a single spin qubit readout in SiC is of high interest to the community, it is not discussed here in terms of new physics but technical improvements, leaving still a quite large number of unanswered questions. As a consequence the paper, as written now, is not suitable for NatureComm.

The photoelectric detection in SiC was reported in the referred publication 10 and the original methodology was developed in previous papers such as those on diamond and properly cited. But, there is no new physics discussed here in a major way (though there is a space for it).

The main problem is as follow (mentioned in the first report). The photoelectric readout in diamond was based on rate dynamics between defect photoionisation from the ES triplet to E_c and the MS transition for spin 1 system with a low spin-orbit interaction. The SiV defect is a $1/2-3/2$ system with two orbital branches. The MS state, on which the photoelectric readout in publication 10 is based upon, is an unconfirmed hypothesis. Could the authors experimentally directly confirm that there is a MS state for this system and how it looks like (symmetry, spin number) ? Then yes the paper would bring new physics

The authors just take over the older hypothesis and don't do efforts to confirm/deny it, for which the single defect centre would be an advantage. For example the authors could execute bunching experiments as a function of the laser power to try to verify it or similar.

Also the authors hypotheses about the 2 photon (and in the rebuttal letter 4 photon for QE – as they claim that one needs still two photons to recharge the state), but the data presented in figure 2f show a quasit linear behavior. This also magnifies the above discussed need to understand the photoionization mechanisms, not taking over only hypothesis from the ref 10. Only indirect measurements based on photocurrent values are provided as a supporting argument for the number of photons needed for one charge carrier pair

Our reply:

First of all, we would like to thank reviewer 2 for their continued engagement with our manuscript and for providing detailed comments during the review process. We sincerely appreciate the time and effort they have devoted to evaluating our work. Their critical perspective has been invaluable in helping us identify areas where clarification and improvement were necessary. Their comments challenged us to refine our explanations and strengthen the manuscript further. While we regret any misunderstandings that may have arisen from our initial responses, we are grateful for the opportunity to address these points more thoroughly.

Based on reviewer 2's comment, we are afraid that our manuscript and the initial response might cause a misunderstanding of the electronic structure of the silicon vacancy centre in silicon carbide. Therefore, we first would like to explain the system. The ground state (GS) of the V2 centre in silicon carbide, the silicon monovacancy at the quasi-cubic site is $S=3/2$ 4A_2 state. The two states denoted as $\pm 3/2$ and $\pm 1/2$ in Fig. 1a are the spin sublevels of the spin quartet state ($m_S=\pm 3/2$ and $m_S=\pm 1/2$), not the orbital branch with spin-orbit interaction. (different from the silicon-vacancy complex defect in diamond). The metastable state (MS) is $S=1/2$, and the existence of the MS of the V2 centre has been intensively discussed in previous theoretical and experimental studies [Refs. 38(37), 55(53), 56(54), Ö. O. Soykal et al., Phys. Rev. B 93, 081207(R) (2016), Ö. O. Soykal et al., Phys. Rev. B 95, 081405 (2017), W. Dong et al., Phys. Rev. B 99, 184102 (2019), H. B. Banks et al., Phys. Rev. Appl. 11, 024013 (2019)], including the power dependence of the bunching experiments [Refs. 38(37) and 55(53)]. Therefore, we believe the research community accepts the existence of the MS state. Further, the spin-dependent intersystem crossing from the excited state (ES) to the MS has also been studied recently in Ref. 56(54), which is the key mechanism for the ODMR and spin polarisation of this defect. Based on these previous findings, we do not discuss the existence of the MS in our manuscript. However, we are afraid that the misunderstanding might be caused by the notations in Fig. 1a, which do not indicate that the states are spin sublevels. Therefore, for readers unfamiliar with the system of the V2 centre, we updated Fig. 1a by adding the label of " m_S " for $\pm 3/2$ and $\pm 1/2$ spin sublevels and denoting the spin number of GS, ES, and MS ($S=3/2$, $3/2$, and $1/2$), respectively.

Even though the existence of the MS and the spin-dependent intersystem crossing mechanism has been shown in previous studies, two important physics have remained a hypothetical proposal in the previous studies (Refs. 10 and 12). The first point is, as pointed out by reviewer 2, the number of photons involved in the photoionisation process. The second point is the dominant ionisation path, whether the ionisation occurs while the system resides in the ES or MS.

Regarding the first point, a quadratic laser power dependence of the photocurrent was reported in Ref. 10. However, the measured photocurrent from the defect ensemble should include the response of defects other than the V2 centre. In our study on single defects, the ionisation is confirmed to be indeed the two-photon process. In the inset of Fig. 2f, we observe a quadratic power dependence of the photocurrent from a single V2 centre at the low laser power region below ~ 1 mW. This quadratic dependence from a single defect is the direct evidence of the involvement of two-step ionisation, as discussed in the NV centre in diamond in Ref. 13. In addition, we quantify the absorption and ionisation rates of the single defect (Supplementary Note 6) from the result of Fig. 2f. This analysis finds that the excitation from the GS to the ES is 50 times faster than the ionisation from the ES, as we discuss in L350–352 and L360–362, which is another new physics we provide in this study. As a result, the photocurrent becomes linear at higher laser powers, which is also observed in the two-photon ionisation for the NV centre in diamond in Ref. 13. Although these rates are determined with continuous-wave excitation (Fig. 2f), we can successfully model the pulsed PDMR dynamics with short laser pulses, reproducing the experimental results (Supplementary Note 7). As reviewer 2 pointed out, the quadratic dependence may not be very clear in Fig. 2f. In fact, in the previous study on the single NV centre (Ref. 13), the quadratic power dependence was not clearly reported. However, in our study, the reduced imaging spot size at the lower power characterised and discussed in Fig. 2g and L182–202 also provides firm evidence of two-photon ionisation, as discussed for the NV centre in diamond in Ref. 14. Importantly, the parameter P_0 , determined from the laser power dependence of the photocurrent (Fig. 2f), well reproduces the behaviour of the spot size reduction in Fig. 2g. Based on these results, we can conclude that the ionisation of the V2 centre is unambiguously the two-photon process. To show these findings, we revised the manuscript by adding explanations on page 7, L172–177 and on page 8, L197–198.

The second point can be discussed from the sign of the spin contrast in PDMR. Suppose the sign of the spin contrast is the same as ODMR. In that case, the ionisation occurs from the excited state because the competitive rate dynamics in the excited state between the ionisation and intersystem crossing in PDMR are similar to those between the radiative decay and the intersystem crossing in ODMR. Conversely, observing the opposite sign of the spin contrast, the metastable state can be identified as the dominant ionisation path [Ref. 41(64)]. In previous studies, the microwave intensity modulation for lock-in detection removed the information of the photocurrent baseline, which makes the spin contrast determination ambiguous. In our study, the lock-in measurement with the laser intensity modulation preserves the information of the photocurrent baseline, making determining the spin contrast sign straightforward. We obtained the same contrast sign in ODMR and PDMR, as shown in Fig. 2e. Additionally, the response of

the spin signal intensity to the laser pulse duration also supports the ionisation from the excited state. If the dominant ionisation path is from the metastable state, the measurement with a short laser pulse may reduce the signal intensity considering the short excited state lifetime of 6 ns and the long metastable state lifetime of $\sim 100\text{--}150$ ns [Refs. 38(37), 55(53)]. However, our study finds that the spin signal intensities measured with a 1- μs -long rectangular 32 mW laser pulse (Fig. 3a) and that with a short 39-ns-long laser pulse at 35 mW (Fig. 3c) are similar. These findings strongly suggest that the dominant ionisation path is from the excited state, providing another new physics. Regarding this point, we revised the manuscript on pages 8–9, L204–215 and page 10, L256. Also, as we identify the ionisation process, we changed the sentences at the beginning of the “Ionisation pulse control for improving single-spin readout efficiency” section (on page 9, L227–231) such that we use confirmed ionisation dynamics rather than mentioning the model as tentative by taking over from the previous study. Additionally, we introduced “MS” as the abbreviation for “metastable state” in this revision process. Therefore, “metastable state” that appears on page 13, L341, is replaced with “MS”.

Reviewer’s comment:

Also, as mentioned in previous review round, there is a basic misunderstanding about the characteristics of back-to-back Schottky diode characteristics and terminology. Flat band conditions on a junction mean, that a voltage is applied equaling the barrier height with an opposite electric field polarity. The authors hypothesize about the barrier height but do not measure it. If it is about the estimated 1.5 V, then the flat band condition (i.e. forward direction) for one of the junction is 1.5 V, it cannot be 9 V or even 10V. At the flat band voltage the contact becomes injecting.

The depletion region is very different terminology, by increasing the voltage on reversely biased junction one depletes the charge from the region between the contact and locate it towards one electrode. I.e. in a p-doped material the holes will be depleted top a width called depletion region width.

The problems that the authors have back-to- back diode, which can be though easily modelled. On the moment when they put flat band voltage on one electrode, the 2nd junction gets double barrier voltage. The location of the defect center in between the contacts and the field profile is utterly important elucidate the transport mechanism. This has not been done.

Although I would be in great favor if single SiV photocurrent technique would be published it

cannot stay solely on technical improvements of previous works and the basic knowledge of device physics on an adequate level needs to be provided. In this sense I did not change mind with respect of the first review. But the potential is there.

Our reply:

We sincerely appreciate reviewer 2 giving us the opportunity to clarify this issue further. The points raised by Reviewer 2 align with the discussions we have provided in our previous response to questions 2-ii-2 and 2-ii-3. We appreciate the opportunity to revisit these points, and we hope the additional clarifications we have now included address any remaining concerns.

We basically agree on the meaning of the flat-band condition and the depletion region for single junctions pointed out by reviewer 2. One thing we would like to note is that the contact is not injecting at the flat-band condition. Here, we define the contact as injecting when the charges are injected from the metal to the semiconductor. At the flat-band voltage, the charge flow from the semiconductor to the metal increases because of the zero-barrier height for charges travelling from the semiconductor to the metal. However, the charge flow from the metal to the semiconductor does not change because the barrier height for electrons travelling from the metal to the semiconductor is unchanged. Instead, the reverse-biased Schottky junction is injecting.

For the metal-semiconductor-metal (MSM) device having two back-to-back junctions, we use the terminology of the “flat-band condition” based on Chapter 7 of Ref. 26 and Supplementary Ref. 1, which is included as Ref. 27 in the revised manuscript. To help readers understand how the MSM Schottky device works for PDMR, we added Fig. 1c, the schematical band diagram of the MSM device biased at the flat-band condition. (In addition, to address the addition of the new Fig. 1c, we added a sentence in the main text on page 5, L111–112.) At the flat-band condition of the MSM device in the definition of Ref. 27, one junction (forward-biased side: junction 2, the right contact in Fig. 1c) achieves the flat band, whereas the other junction (reverse-biased side: junction 1, the left contact in Fig. 1c) has a bent potential. The process to achieve the flat band at junction 2 is discussed in detail in Ref. 27. Below, we explain this process by following this reference. As reviewer 2 pointed out, the forward-biased junction achieves a flat band if *the voltage across the junction*, V_2 , is larger than the Schottky barrier height (more precisely, $V_2 > V_{bi}$: built-in potential). However, in MSM devices, it is important to note that the voltage is applied not directly across the junction but across the two electrodes: $V=V_1+V_2$, where V is the voltage applied to the device, and V_1 and V_2 are voltages across the reverse-biased and forward-biased junctions, respectively. As we already pointed out in the previous response letter, when a small bias voltage V is applied, this voltage is mostly consumed by the reverse-biased junction V_1 that expands the depletion layer,

as discussed in Ref. 27. This is because of the current continuity condition at the two junctions: $J_1(V_1) = J_2(V_2)$. As junction 2 is forward-biased, J_2 can increase in large amounts even with a small change of $|V_2|$. On the other hand, the change in J_1 by $|V_1|$ is small because junction 1 is reverse-biased. For this reason, applying $V = 1.5$ V does not achieve the flat-band condition at junction 2. By increasing the applied voltage, the depletion layers of the two junctions contact each other (fully-depleted condition, or reach-through condition in Ref. 27). Further increase in the applied voltage reduces the potential barrier at the forward junction, achieving the flat band at junction 2. The voltage required to achieve this condition is described by $V_{\text{FB}} = eN_A L^2 / 2\epsilon_s$ as we wrote in Supplementary Information 1 by citing Supplementary Ref. 1, giving 9.1 V in the case of our device.

Reviewer #3 (Remarks to the Author):

Reviewer's comment:

The authors have satisfactorily responded to my comments and questions. They explained that the much higher conductivity of SiC presents unique challenges, creating large background currents that would saturate any high gain amplifier and swamp the tiny photocurrent from single defects. Even if optical lock in was implemented, this would make single defect sensing impossible. This motivates the specific use of Schottky type contacts, and this is one of the things that enables single defect PC measurements. This point is made in the revised manuscript and I think justifies the paper's claims to significance.

I thank the authors for their detailed comments and additional measurements, and note that the addition of Ramsey and Spin echo data really pushes the paper over the line as far as coherent spin control goes. The authors changes to the paper are extensive and thorough, I am happy to recommend publication in Nat Comms.

Our reply:

We are extremely grateful for reviewer 3's thorough review and kind comments on our revised manuscript. We deeply appreciate reviewer 3's acknowledgement of the significance of the unique challenge in silicon carbide and the added measurement data. The detailed feedback has been invaluable in improving the clarity and impact of our work, and we are truly grateful for reviewer 3's time and effort in helping refine the manuscript. We thank reviewer 3 for the recommendation for publication.

Revision outside the comments from the Reviewers:

(A) We revised a redundant sentence on page 6, L126–127.

Before: “Furthermore, to further suppress the leakage current, the Schottky contact area is designed to be relatively small (approximately $120 \mu\text{m}^2$) to minimize the leakage current.”

After: “Furthermore, to minimise the leakage current, the Schottky contact area is designed to be relatively small (approximately $120 \mu\text{m}^2$). ”

(B) Ref. 62(60)’s volume and page information was added instead of DOI.

(C) We corrected a typographical error on page 14, L366: from “Fig. 1d” to “Fig. 2d”.

(D) We corrected a typographical error on page 30, L737, in the figure caption of Fig. 3g: from “the0” to “the”.

Point-by-point response to the REVIEWER COMMENTS

Reviewer #1 (Remarks to the Author):

Reviewer's comment:

We appreciate that the authors have provided satisfactory responses to all of our questions and concerns during the first and second rounds of review. We are pleased to recommend this paper for publication in Nature Communications.

To enhance the impact of the current work, I encourage the authors to provide additional details regarding the methods they employed. Below are some specific suggestions for the authors' consideration:

Our reply:

We sincerely appreciate Reviewer 1's constructive feedback throughout the review process and their acknowledgment of our revised manuscript. We are also grateful for the recommendation for publication and for the encouragement to further enhance our work. In response to the three comments regarding additional details on our technical methods, we address each point in detail below.

Reviewer's comment:

1. As the authors note, there is small frequency-dependent crosstalk from the RF signal to the detected photocurrent. Since the dynamics of carriers in semiconductors can be influenced by the electromagnetic field of the RF signal, the crosstalk can induce significant noise during PDMR measurements. When using lock-in amplification, the RF amplitude is modulated at the modulation frequency, and the crosstalk can manifest as a large noise signal oscillating at the same frequency. This could severely compromise the measurement. Based on the excellent results presented by the authors, it appears that RF crosstalk is nearly entirely suppressed in this experiment. Could the authors provide further details on the methods they used to suppress RF crosstalk noise?

Our reply:

While we appreciate the reviewer's positive assessment, we would like to clarify that our experiment did not involve additional measures to suppress RF crosstalk beyond what was achieved in previous studies. As the reviewer highlighted, and as we mentioned in L223, there is a small RF frequency-dependent crosstalk to the detection electronics. The frequency-dependent background in the RF-swept PDMR (Fig. 2e, shown after background subtraction as described in the figure caption) is on the order of ~ 1 fA over a 10 MHz range, which is comparable with previously reported PDMR measurements of the spin ensemble of the V2 centre (Ref. 12). At present, we are unable to further

suppress the RF crosstalk beyond what was achieved in previous studies. Addressing the limitation remains a future task. However, we note that the crosstalk-induced electrical current remains constant when both the RF frequency and pulse length are fixed. With the constant crosstalk, the baseline is not affected, and the crosstalk does not appear as a noise in the spectrum. For this reason, we employed the magnetic-field sweep technique instead of the RF-frequency sweep for Figs. 3a–c, as described in L222–224 of the main text. This method has been previously employed in Ref. 10. Therefore, we specifically address the effect of magnetic-field sweep in the last sentence of Section “Results–Photoelectrical imaging and spin detection of single defect in silicon carbide” (L222–225 in the revised manuscript):

Previous version: “To focus on the laser effect, we measure PDMR with a magnetic field sweep in the experiments below as we observe RF frequency-dependent small crosstalk to the detection electronics.”

Revised version: “However, we observe RF frequency-dependent small crosstalk to the detection electronics. To focus on the laser effect in the experiments below, we measure PDMR with a magnetic field sweep, such that the current component from the RF crosstalk remains constant¹⁰.”

Reviewer’s comment:

2. In Figure 1b, the authors provide a side view of the sample and its layered structure. However, the shape and relative positioning of the electrodes and the RF antenna are also critical for efficient photocurrent collection and minimizing RF noise. We recommend that the authors include a schematic diagram showing the vertical view of the fabricated sample surface, illustrating the complete configuration of the electrodes and the RF antenna.

Our reply:

We thank the reviewers for the comments. In accordance with the reviewer’s recommendation, we have added a schematic diagram illustrating the vertical view of our device at the bottom right of Fig. 1b.

Reviewer’s comment:

3. As described by the authors, the sample is mounted and wire-bonded onto a custom PCB. Could the authors elaborate on the grounding methods employed in the design of the PCB? Specifically, do the RF circuit, the bias voltage application circuit, and the photocurrent collection circuit share a common ground? The grounding approach is critical for effective noise suppression.

Our reply:

We sincerely thank the reviewer 1 for the question. Our PCB is an in-house fabricated microstrip line structure for RF, with its ground plane electrically connected to the RF ground and the signal (bias voltage and photocurrent collection) ground. To explain the PCB and grounding, we changed the last paragraph of “Sample preparation” section in Methods:

Previous version: “... and wire bonded on a hand-made printed circuit board.”

Revised version: “... and wire bonded onto an in-house fabricated printed circuit board with a microstrip line structure for RF, with its ground plane electrically connected to both the RF line ground and the signal line ground”.

Minor revisions outside of the reviewers’ comments

1. During the preparation of the publication-quality figures, we carefully reviewed our data and identified a minor inconsistency between the numerical values and the plots presented in the original figures. We have now corrected these discrepancies in the final manuscript and figures, which has resulted in small modifications to the signal-to-noise ratio (SNR) data in the plot of ODMR-C in Fig. 4b, the laser power values for the PDMR data in Fig. 4c, and the laser power values of the experimental data (Exp.1 and Exp.2) in Supplementary Figs. 6c–f. Additionally, to maintain consistency with the revised figures, some laser power values described in the main text (in L254, L728, and L760) have been updated accordingly. Also, we identified in Fig. 3c that the length of the inset bar corresponding to the signal contrast of 1% was displayed slightly shorter than we had intended. In the submitted figure, we have corrected the length of the bar. We would like to emphasize that these corrections do not affect the discussion and conclusions of this study.
2. In Fig. 1a, we slightly modified the line shape of the yellow broken arrows so that the readers can more easily recognize that these lines are the broken lines denoted in the figure caption.
3. After the first revision, Figure 3 includes the contents of the coherent spin readout. Therefore, we have added “... and coherent spin readout” at the end of the figure title.